# DNA methylation at birth in monozygotic twins discordant for pediatric acute lymphoblastic leukemia

Eric M. Nickels [1,2], Shaobo Li [2], Swe Swe Myint [2], Katti Arroyo[2], Qianxi Feng[2], Kimberly D. Siegmund [2], Adam J. de Smith [2] & Joseph L. Wiemels [2] ✉

Aberrant DNA methylation constitutes a key feature of pediatric acute lymphoblastic leukemia at diagnosis, however its role as a predisposing or early contributor to leukemia development remains unknown. Here, we evaluate DNA methylation at birth in 41 leukemia-discordant monozygotic twin pairs using the Illumina EPIC array on archived neonatal blood spots to identify epigenetic variation associated with development of pediatric acute lymphoblastic leukemia, independent of genetic influence. Through conditional logistic regression we identify 240 significant probes and 10 regions associated with the discordant onset of leukemia. We identify a significant negative coefficient bias, indicating DNA hypomethylation in cases, across the array and enhanced in open sea, shelf/shore, and gene body regions compared to promoter and CpG island regions. Here, we show an association between global DNA hypomethylation and future development of pediatric acute lymphoblastic leukemia across disease-discordant genetically identical twins, implying DNA hypomethylation may contribute more generally to leukemia risk.

Acute lymphoblastic leukemia (ALL) is the most common malignancy in childhood, accounting for 25–30% of all pediatric cancer diagnoses[1]. Despite overall survival >90%[2], pediatric ALL remains a leading cause of pediatric cancer-related morbidity and mortality[3], with chemotherapy leading to significant acute and long-term toxicities for survivors[4]. While the etiology of pediatric ALL is not fully elucidated, the median age of onset between 4 and 5 and the presence of disease-defining chromosomal translocations at the birth point to an *in utero* origin[5]. Monozygotic twin studies show a concordance rate of ~10%, in which twin pairs harbor the same initiating chromosomal translocation through shared blood chimerism[6]. However, given a discordance rate of 90%, these translocations alone are not sufficient for the development of ALL, indicating additional intrauterine or early life genetic, epigenetic, and environmental factors contribute as necessary second hits for leukemia development. This includes a potential role for DNA methylation as an early contributor or predisposing factor in ALL development.

DNA methylation is a mitotically heritable, stable epigenetic marker largely established in early embryogenesis under the influence of genetic, environmental, and stochastic (or random) control[7]. The periconceptional period reveals a crucial window on the establishment of DNA methylation patterns, which may persist lifelong, with the potential to influence phenotypic expression later in life. This includes a monozygotic twin-specific DNA methylation signature established early in embryogenesis which persists into adulthood[8]. Several intrauterine factors, including the availability of maternal methyl-donor nutrients such as folate, are known to influence the way DNA methylation is established during this period[9]. For this reason, DNA methylation has been considered a potential mechanism mediating the relationship between maternal exposures and the risk of pediatric ALL[10]. Importantly, exposure to nutrient availability and other factors are not necessarily shared equally in monozygotic twin gestations due to unequal sharing of placental blood flow[11], creating a potential imbalance in disease risk in otherwise genetically identical individuals.

[1]Children's Hospital Los Angeles, Cancer and Blood Disease Institute, Los Angeles, CA, USA. [2]Center for Genetic Epidemiology, Department of Population and Public Health Sciences, Norris Comprehensive Cancer Center, University of Southern California, Los Angeles, CA, USA. ✉e-mail: wiemels@usc.edu

Epigenetic variation has been associated with phenotypic variation between otherwise genetically identical monozygotic twin pairs, including the discordant onset of disease[12].

Aberrant DNA methylation is a hallmark of ALL at diagnosis[13-16]. Global DNA hypermethylation, specifically located in CpG island and promoter regions has been demonstrated in ALL cells compared to healthy control bone marrow samples[13,15]. In comparison to normal B-cell precursor cells, epigenetic remodeling in pediatric B-cell ALL demonstrates de novo DNA methylation in small functional compartments such as CpG islands and promoters, while DNA demethylation occurred in large intercompartmental backbones, such as repetitive regions in the genome[17]. In addition, subtype-specific DNA methylation signatures have been identified[14], which can be used in a predictive manner to identify pediatric ALL by cytogenetic subtype[18]. A subset of this CpGs is shared amongst all ALL subtypes, constituting a core set of aberrant sites of DNA methylation throughout the genome[14,15]. While the involvement of DNA methylation at the time of diagnosis is well established, its role as a predisposing or early contributor to the development of ALL has not been reported.

DNA methylation is most frequently established in a cell-type specific manner; however, ~0.1% of the epigenome is concordant across all tissues in an individual. DNA methylation at these sites of correlated interindividual variation (CoRSIV) is sensitive to the influence of the periconceptional intrauterine environment[19,20], with suggested influence from maternal nutritional status. CoRSIV sites have the potential to function as metastable epialleles, generating phenotypic variation between individuals independent of genetic influence[7,20]. These sites of interindividual variation describe a mechanism by which environmentally-sensitive establishment of DNA methylation patterns at birth can moderate disease risk[20].

Owing to their genetic identity, discordant monozygotic twins, in which one twin develops a disease and the other does not, provide an ideal setting to investigate the role of epigenetic influence on disease risk[12]. We hypothesized that variation in DNA methylation at birth, as a reflection of unequal sharing of the intrauterine environment, contributes to the differential risk of leukemia between discordant monozygotic twins. We utilized archived neonatal blood spots from discordant monozygotic twin pairs to investigate this relationship with the goal of identifying sites of DNA methylation uniquely associated with the future development of ALL.

In this work, we show a significant association between DNA methylation variation in identical twins at CpG sites and regions across the epigenome and the discordant future development of ALL using a conditional regression model. This includes a total of 240 significant differentially methylated CpGs and 10 regions across the epigenome associated with the future onset of ALL. We further describe significant global DNA hypomethylation in ALL cases compared to their matched twin sibling controls. Furthermore, the degree of DNA hypomethylation is higher in the open sea and gene body regions of the genome compared to CpG islands and promoter regions. These results imply DNA hypomethylation may contribute more generally to ALL risk.

## Results

### Subject characteristics

Genome-wide DNA methylation data were obtained for 43 ALL-discordant monozygotic twin pairs (43 ALL cases and 43 unaffected siblings) using the Illumina Infinium Methylation EPIC BeadChip array. Characteristics of these twin pairs are shown in Table 1, with information derived from the California Cancer and Vital Statistics registries. The median gestational age was 258 days (36 weeks), ranging from 184 days (26 weeks) to 306 days (43 weeks). No significant difference was noted in birthweight between cases and unaffected siblings ($P = 0.17$ by two-sided paired $T$ test). The age of diagnosis in the case twin ranged from <1 to 23 years (median = 5). A larger proportion of twin pairs were female, likely due to sampling bias. Diagnoses included precursor cell lymphoblastic leukemia, not otherwise specified (NOS, $n = 19$), B-lymphoblastic leukemia/lymphoma, NOS ($n = 14$), precursor B-cell lymphoblastic leukemia ($n = 7$), T-lymphoblastic leukemia/lymphoma ($n = 2$), and leukemia/lymphoma with t(12;21) (p13;q22);TEL-AML1(ETV6-RUNX1) ($n = 1$). Of these, 32 were denoted B-cell and 4 T-cell lineages, while 7 did not have a listed cell lineage. Following DNA methylation array quality control and normalization, two twin pairs were removed due to significantly elevated mean detection $P$-values. This resulted in a final set of 710,010 array probes passing quality control measures among the 41 twin pairs included in further analysis. Cell proportions were subsequently compared across twin pairs between cases and unaffected siblings using a paired Wilcoxon signed-rank test, which showed no significant differences in nucleated cell proportions (Supplementary Fig. 1, Supplementary Data 1). Correlation in beta values between twin pairs ranged from $R = 0.968$ to $0.991$ (Spearman) for all 710,010 probes (Supplementary Fig. 2). tSNE analysis, omitting chromosomes X and Y to minimize sex-related influences, shows a strong association between related twin pairs, with no obvious clustering by EPIC array chip suggesting bias from batch effect (Supplementary Fig. 3).

### Within-pair assessment

To assess absolute DNA methylation differences across individual twin pairs, we conducted a within-pair assessment evaluating delta beta (case $\beta$ value minus control $\beta$ value) values across all 710,010 array probes. Probes meeting a threshold absolute delta beta value difference of 0.15 or greater were included in the analysis as sufficiently variable between twins. We identified a total of 18,001 probes across the 41 twin pairs meeting the 0.15 threshold in absolute *delta beta*

## Table 1 | Subject characteristics

| | | | |
|---|---|---|---|
| Sex | Female | 27 (63.8%) | |
| | Male | 16 (37.2%) | |
| Race/ethnicity | Non-Hispanic White | 11 (27%) | |
| | Non-Hispanic Black | 3 (7%) | |
| | Hispanic | 24 (59%) | |
| | Asian-Pacific Islander | 3 (7%) | |
| Median gestational age (range) | | 258 days (184–306) | |
| Birthweight (mean) | Case twin | 2427.4 g | $P = 0.17^1$ |
| | Unaffected sibling | 2363.3 g | |
| Age of diagnosis (years) | | 0–23 (median 5) | |
| Diagnosis | Precursor cell lympho-blastic leukemia, NOS | 19 (44.1%) | |
| | B-lymphoblastic leukemia/lymphoma, NOS | 14 (32.6%) | |
| | Precursor B-cell lympho-blastic leukemia | 7 (16.3%) | |
| | T-lymphoblastic leukemia/lymphoma | 2 (4.7%) | |
| | Leukemia/lymphoma with t(12;21)(p13;q22);TEL-AML1(ETV6-RUNX1) | 1 (2.3%) | |
| Cell Lineage | B-cell | 32 | |
| | T-cell | 4 | |
| | Not listed | 7 | |

Birth and diagnostic features of the 43 twin pairs who underwent DNA methylation array analysis are demonstrated in Table.
[1]Two-sided paired Wilcoxon test. *NOS* not otherwise specified.

variation. A total of 3937 probes located within 297 genes were recurrently variable, meeting the 0.15 difference in absolute delta beta value threshold in at least two separate twin pairs. Gene set enrichment analysis was conducted on the 3937 recurrently variable probes (Supplementary Data 2). This resulted in 573 gene ontology terms with $P < 0.05$, with 7 of the top 15 terms linked to immune-related processes (Supplementary Data 3). No ontology terms were significant after correction for multiple comparisons. Similarly, 4 of the top 15 KEGG-pathway terms, including the top term "T-cell receptor signaling pathway," were immune-related with nominal $P < 0.05$, however, these were not significant after correction for multiple comparisons (Supplementary Data 4).

## Conditional regression assessment

We next utilized a conditional regression model assessing the relationship between DNA methylation at all 710,010 array CpGs and leukemia status accounting for the paired nature of the data set while controlling for batch effects and nucleated cell proportions obtained from DNA methylation-supervised cell deconvolution analysis. Conditional regression analysis was conducted on B-cell and unknown lineage cases and unaffected siblings ($n = 37$ pairs) to focus on ALL cases with the most similar assumed underlying pathophysiology. T-cell cases ($n = 4$) were not analyzed separately, as the small sample size precluded adequate assessment by the regression model. Consistent results with our findings presented here were generated in models including all 41 pairs, as well as with the 30 confirmed B-ALL cases (removing T-cell and those that are NOS but assumed to be majority B-cell). Conditional regression analysis resulted in a total of 240 differentially methylated probes (DMPs) meeting a threshold of FDR $< 0.05$ (Table 2, Fig. 1a), with a Q-Q plot demonstrating minimal genomic inflation with $\lambda = 1.02$ in Fig. 1b. Full regression results including mean beta values are listed in Supplementary Data 5. Plots demonstrating case and control beta values by twin pair in the top 20 most significant DMPs are shown in Supplementary Fig. 4. Of the significant DMPs, 3 overlapped with probes identified as harboring constitutional differential DNA methylation in ALL at diagnosis[14]. An additional 17 DMPs were in genes identified to be differentially methylated in the same study, creating overlaps in *RUNDC3B, ABCB1, MARVELD3, SORCS3, FEZF1, PRDM16, ANK1, CNTNAP5, PREX2, DSCAM, ARHGEF4, SYT13, ZNF274, TBX4, NELL2, ADAMTS16, CAMTA1, OSR1, RXRB*, and *SNX31*. Gene set enrichment analysis of the 240 significant DMPs did not identify significantly enriched gene ontology or KEGG-pathway terms (Supplementary Data 6, 7). We next evaluated differentially methylated regions (DMRs) using comb-p[21], which identifies significant DMRs through spatial correlation of $P$-values obtained from conditional regression analysis. We identified 10 significant DMRs with Šidák $P < 0.05$ (Table 3, Fig. 1c), which accounts for multiple comparisons in the comb-p model. Of these, 7 demonstrated a consistent direction of effect in coefficients for CpG probes within the DMR (Supplementary Data 8). The most significant region encompassed a 454 bp region on chromosome 6 associated with *TRIM39-RPP21* (Šidák $P = 2.39 \times 10^{-9}$, Table 3, Supplementary Fig. 5). A separate region associated with *AMH* overlapped with a CoRSIV region (Šidák $P = 0.007$) and was also previously described as differentially methylated in ALL at diagnosis[14]. Regional gene set enrichment analysis did not identify any significantly enriched gene ontology or KEGG-pathway terms (Supplementary Data 9, 10).

## Validation of DNA-methylation results

To validate findings from the EPIC array data, we performed methylation-specific droplet digital PCR (ddPCR) to evaluate the DNA methylation status of four significant DMPs with the highest intra-pair variability in DNA methylation (at *TRIM39, FOXK1, CMIP*, and *SDHC*) using nine twin pairs ($n = 18$ individuals) with sufficient remaining

genomic DNA (Supplementary Data 11). Normalized DNA methylation status from array data (beta values) was compared to ddPCR (fractional abundance, or the proportion of positive methylated droplets divided by total positive droplets) results for each target DMP (Supplementary Data 12). There was significant correlation between the two methods (Pearson $R = 0.81$, $P < 2.2 \times 10^{-16}$) for all individuals across the four DMPs (Supplementary Fig. 6a). Each target DMPs remained significant when assessed individually, including cg17080697 at *TRIM39* ($R = 0.94$, $P = 1.0 \times 10^{-08}$), cg14562331 at *CMIP* ($R = 0.87$, $P = 2.9 \times 10^{-06}$), cg04976226 at *FOXK1* ($R = 0.81$, $P = 5.1 \times 10^{-05}$), and cg11744295 at *SDHC* ($R = 0.63$, $P = 0.005$). Comparison of normalized delta-DNA methylation values (case minus control DNA methylation) for the two methods were also significantly correlated for all four target DMPs ($R = 0.56$, $P = 0.00013$) (Supplementary Fig. 6b). Individually, all target DMPs remained positively correlated, with one of four targets remaining significantly correlated including *TRIM39* ($R = 0.68$, $P = 0.044$), *CMIP* ($R = 0.52$, $P = 0.15$), *FOXK1* ($R = 0.66$, $P = 0.055$), and *SDHC* ($R = 0.52$, $P = 0.15$). For all normalized delta-DNA methylation comparisons for the nine twin pairs across four separate DMP targets, 72% (26/36) showed a concordant direction of effect (binomial test $P = 0.011$).

## Assessment of DNA hypomethylation by genomic region

Global DNA methylation content was significantly reduced in the case of twins compared to controls (Paired Wilcoxon Test $P = 0.048$, Fig. 2a, Supplementary Data 13). A single twin pair (Pair 14) had notably overall lower global DNA methylation content than all other pairs (Fig. 2a); this pregnancy was the only twin pair diagnosed with gestational diabetes. The same pair was also notably higher in nucleated red blood cell content compared to all other pairs (Supplementary Fig. 1). To better understand how DNA hypomethylation was distributed across genomic regions, we next evaluated coefficient direction from the conditional regression analysis by genomic and epigenomic context. Across all array, probes compared between the case twins and their unaffected siblings, a significant bias in the frequency of negative regression coefficients was identified (Fig. 2b, Table 4), with 409,819 (57.7%) demonstrating negative coefficients compared to 300,191 positive coefficients (binomial test $P = 2 \times 10^{-323}$). This trend was consistent among the 240 significant probes, where 157 (65.4%) demonstrated negative coefficients. We further assessed this bias by genomic regions annotated in our conditional regression analysis using the Illumina Epic array annotations. Based on the relationship to CpG island (CGI), 59.0% of probes associated with open sea regions ($n = 228,222$ of 387,108 total probes) and 58.7% of probes in CGI shelf/shore regions ($n = 52,579$ of 89,500 total probes) were more highly enriched in negative coefficients compared to the array overall, while just 53.3% of island probes ($n = 76,982$ of 144,560 total probes) were associated with negative coefficients, notably less than the array overall. When assessing associations based on the Regulatory Feature Group, 60.1% of probes within genes ($n = 1253$ of 2086 total probes) had negative coefficients, while only 52.0% of probes within promoter regions ($n = 27,523$ of 52,939 total) had negative coefficients. Consistent with this finding, when assessed by UCSC RefGene Group, 56.7% of TSS1500 probes ($n = 50,787$ of 89,575 total) and 53.0% of TSS200 probes ($n = 30,824$ of 58,150) had negative coefficients, which is lower than the percentage seen in the full array. To further confirm the regression coefficient bias by region, we assessed conditional regression results in both the full cohort including T-ALL cases ($n = 41$ twin pairs) and B-cell cases alone ($n = 30$ pairs). Both groups demonstrated similar negative coefficient bias across array probes in an equivalent pattern the $n = 37$ twin pairs show.

As negative coefficients in the conditional regression model indicate a relationship between hypomethylation in ALL cases compared to controls, we next sought to confirm whether hypomethylation is similarly identified in these regions based on raw DNA

**Table 2 | Top differentially methylated probes from conditional regression analysis**

| Chr. | Position (hg19) | CpG name | UCSC RefGene name | Relation to Island region | Promoter associated | Mean beta (SD) | Mean delta-beta (SD) | Regression results | |
|------|-----------------|----------|-------------------|---------------------------|---------------------|----------------|----------------------|--------------------|--------|
| | | | | | | | | Coef. | FDR |
| 14 | 73976155 | cg07122798 | *HEATR4* | Open sea | No | 0.920 (0.025) | −0.00675 (0.0144) | −2.954 | 2.22E-05 |
| 19 | 51375975 | cg19582822 | *KLK2* | Open sea | No | 0.852 (0.045) | 0.0101 (0.0156) | 1.833 | 1.83E-04 |
| 1 | 24104752 | cg22131571 | *C1orf128* | Island | Yes | 0.069 (0.008) | −0.00512 (0.00873) | −2.918 | 1.83E-04 |
| 15 | 90447350 | cg10093558 | *ARPIN; C15orf38-AP3S2* | Open sea | No | 0.942 (0.018) | 0.00782 (0.0148) | 2.029 | 1.83E-04 |
| 21 | 32649215 | cg05299823 | *TIAM1* | Open sea | No | 0.944 (0.029) | 0.00248 (0.00663) | 3.157 | 6.19E-04 |
| 12 | 27462965 | cg22709041 | *STK38L* | Open sea | No | 0.919 (0.014) | 0.00511 (0.0117) | 2.590 | 6.56E-04 |
| 2 | 111493356 | cg09756855 | *ACOXL* | Shelf | No | 0.666 (0.027) | −0.0128 (0.0231) | −3.873 | 6.77E-04 |
| 1 | 27114177 | cg12806381 | *PIGV* | Island | No | 0.071 (0.011) | 0.00734 (0.0123) | 2.036 | 6.77E-04 |
| 9 | 2717531 | cg26428825 | *KCNV2* | Shore | No | 0.950 (0.015) | −0.00600 (0.0107) | −2.503 | 0.00217 |
| 17 | 79479583 | cg05984533 | *ACTG1* | Island | No | 0.029 (0.006) | 0.00336 (0.00718) | 1.698 | 0.00217 |
| 1 | 101361740 | cg16449184 | *EXTL2; SLC30A7* | Island | Yes | 0.053 (0.009) | −0.00564 (0.0117) | −2.446 | 0.00306 |
| 6 | 31934523 | cg06250213 | *SKIV2L* | Open sea | No | 0.914 (0.014) | 0.00738 (0.0172) | 2.212 | 0.00318 |
| 2 | 170682238 | cg23927974 | *METTL5* | Shore | No | 0.927 (0.012) | −0.00970 (0.0157) | −2.352 | 0.00331 |
| 21 | 43240484 | cg14282798 | *PRDM15* | Island | No | 0.904 (0.015) | −0.0120 (0.0187) | −2.327 | 0.00362 |
| 19 | 14089711 | cg18642503 | *RFX1* | Island | No | 0.056 (0.006) | −0.00349 (0.00656) | −3.349 | 0.00389 |
| 20 | 32700182 | cg19881050 | *EIF2S2* | Island | Yes | 0.036 (0.006) | 0.00430 (0.00814) | 1.721 | 0.00425 |
| 5 | 131826485 | cg06942904 | *IRF1* | Island | Yes | 0.066 (0.013) | 0.00964 (0.0189) | 1.901 | 0.00486 |
| 2 | 74919244 | cg05363574 | Intergenic | Open sea | No | 0.955 (0.015) | −0.00366 (0.00620) | −3.168 | 0.00486 |
| 7 | 105679949 | cg06475386 | Intergenic | Open sea | No | 0.940 (0.013) | 0.00418 (0.0106) | 3.164 | 0.00486 |
| 11 | 32915131 | cg05662684 | *QSER1* | Island | Yes | 0.078 (0.007) | −0.00338 (0.00885) | −3.647 | 0.00486 |

Conditional logistic regression analysis was used to assess for a relationship between leukemia status and DNA methylation at the 710,010 array probes included in the study, controlling for sex, array plate, nucleated cell proportions, and clustering by twin pair identity. The top 20 differentially methylated probes (DMPs) ranked by FDR-corrected conditional regression *P* value are shown out of a total of 240 significant (FDR < 0.05) DMPs. Probes without a genetic association are denoted as intergenic. Mean beta and delta-beta values are shown for twin pairs included in the conditional regression analysis (n = 37). *Chr* Chromosome. *Coef* Coefficient. *FDR* False discovery rate. *SD* Standard deviation. *UCSC* University of California Santa Cruz.

methylation beta values. Using the same probe associations as with our coefficient analysis, we obtained median delta beta values by pair by genomic region among the full set of 41 discordant twins (Fig. 2c, d, Table 4). Across the full set of 710,010 array probes, a significant shift toward DNA hypomethylation was noted in ALL cases compared to sibling controls (Wilcoxon signed-rank *P* = 0.009). A consistent significant trend toward DNA hypomethylation was identified in probes associated with 3′-UTR, 5′-UTR, gene body, CGI shelf/shore, open sea, and TSS1500 regions (raw *P* and FDR-corrected *P* < 0.05). However, median delta beta values were not significantly different from zero in the island, promoter, and TSS200 regions, and trended toward DNA hypermethylation in cases compared to controls. Twin pairs demonstrating global DNA hypomethylation in the case twin (n = 30) also tended to show hypomethylation specifically in open sea regions (n = 28), however, these same twin pairs showed median island values near zero or, in some cases, showing hypermethylation in cases (Fig. 2e, f). A similar finding was identified by evaluating median values for promoter regions versus gene body regions (Fig. 2g, h), indicating regional specificity of the DNA hypomethylation profile identified in this study. These 30 hypomethylated twin pairs did not differ significantly from the remaining 11 twin pairs regarding the age of leukemia diagnosis, diagnosis code, birthweight, or array chip/batch number.

**Assessment of DNA methylation in repetitive elements**
Given the bias towards DNA hypomethylation in cases in open sea regions, which are enriched in repetitive elements, we next assessed the specificity of the coefficient bias by type of repeat. Using the UCSC Genome Browser RepeatMasker track we cataloged positional overlaps with the 710,010 array probes. A total of 336,695 probes overlapped with repetitive elements, of which 193,966 were associated with negative coefficients in the regression analysis (Binomial test $P = 2 \times 10^{-323}$). Of the 19 classes of repetitive elements annotated in the UCSC Genome Browser data, 11 demonstrated a significant negative bias, with the strongest bias noted in LINE (n = 72,266 of 125,087 total probes, FDR-corrected $P = 9 \times 10^{-323}$) and SINE associated probes (n = 68,984 of 119,935 probes, FDR = $9 \times 10^{-323}$) Supplementary Fig. 7a, Supplementary Data 14). This trend was similar in CpGs overlapping repetitive elements within open sea regions (n = 187,001 CpGs, 58.9% negative coefficients) and CpG sites not associated with repetitive elements in open sea regions (n = 200,555 CpGs, 59.0% negative coefficients, *P* = 0.421). Median delta beta values were significantly associated with DNA hypomethylation in ALL cases in 18 of 19 repetitive element classes (FDR < 0.05, Wilcoxon signed-rank test, Supplementary Fig. 7b, c, Supplementary Data 14).

**Assessment of DNA methylation by transcription factor binding motif**
We next assessed for enrichment in transcription factor (TF) binding motif overlaps with the 240 significant probes identified in the conditional regression analysis. We utilized the genomic positions of 148 TF-binding motifs annotated in the ENCODE ChIP-seq database to identify 2,516,987 overlaps with 317,558 distinct probes on the full array. In comparison, we identified a total of 1057 overlaps with 103 of 240 significant probes from the regression analysis and 127 TF-binding motifs. A total of 6 TF-motifs demonstrated raw *P* < 0.05 for enrichment in significant probes, including ATF3, E2F5_(H-50), SIX5, SMC3_(ab9263), p300_(F-4), and USF-1. None was significantly enriched after FDR correction for multiple comparisons (Supplementary Data 15).

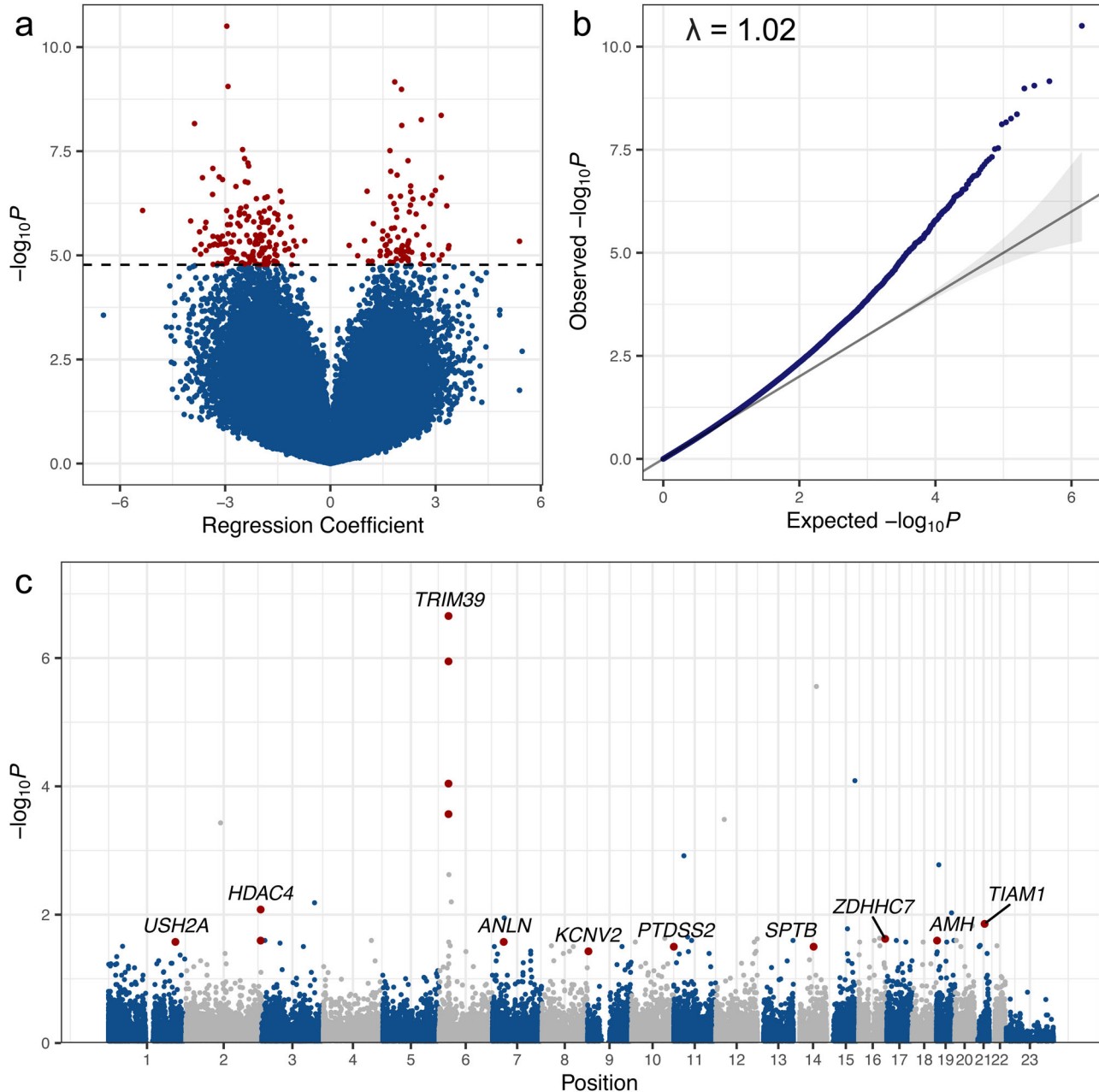

**Fig. 1 | Conditional regression analysis of 37 twin pairs discordant for ALL. a** Volcano plot demonstrating the distribution of coefficients and $-\log_{10}P$ values for 710,010 probes assessed from the DNA methylation array. Coefficients and $P$ values were calculated using conditional logistic regression to test the association of DNA methylation at each CpG with ALL development, adjusting for sex, array plate, nucleated cell proportions, and clustering by paired twin relationships. Dotted line indicates the threshold for false discovery rate (FDR) adjusted significance. Significant probes (FDR < 0.05, $n = 240$) are highlighted in red. **b** Q–Q plot demonstrating observed versus expected $P$ values across the array. Genomic inflation is represented by $\lambda = 1.02$. Trend line represents observed $-\log_{10}(P)$ = expected $-\log_{10}(P)$. Gray-shaded area represents 95% confidence interval. **c** Manhattan plot of regionally adjusted $P$ values showing the genomic location of the 10 significant differentially methylated regions (DMRs) identified in regional analysis. Significant regions (shown in red) were defined as Šidák-corrected $P < 0.05$ identified using comb-p. The most significant DMR, associated with *TRIM39-RPP21*, includes a 454 bp region encompassing 9 probes on chromosome 6 (note only four significant probes located within this region are shown due to overlapping $P$ values). A second 167 bp region on chromosome 19 including four probes associated with *AMH* overlaps with a CoRSIV site. Source data are provided as a Source data file.

A significant negative regression coefficient bias was identified in 87 of the 148 TF-motifs, while a single TF-motif (SIX5) had a significant positive bias (Supplementary Data 16). Median delta beta values were more likely to be positive (104 of 148 TF-motifs), however, there were no significantly shifted values for any TF-motif after FDR correction for multiple comparisons. There were 8 TF motifs when analyzed with a more generous FDR cutoff (<0.01), including motifs for BRF2, c-Jun, and STAT3 (Supplementary Fig. 8).

## Assessment of CoRSIV overlaps by regulatory elements

We next looked at the 1128 array probes overlapping 756 distinct regions defined as CoRSIVs (correlated regions of systemic inter-individual variation, $n = 10,672$ total regions across the genome, Supplementary Data S17). Of the CoRSIV overlapping probes, 630 (55.9%) had negative coefficients (Supplementary Fig. 9a), which is significantly biased by the binomial test (FDR-adjusted $P = 9.44 \times 10^{-5}$), however, this is similar to the distribution of negative coefficients

**Table 3 | Differentially methylated regions**

| Chr | Start (hg19) | End (hg19) | Width | # Probes | Coefficient Direction | Mean Region Beta (SD) | Mean Region Delta-Beta (SD) | Šidák P value | Associated gene | Promoter associated | CoRSIV overlap |
|---|---|---|---|---|---|---|---|---|---|---|---|
| 6 | 30297174 | 30297627 | 454 | 9 | - - - - - - - - - | 0.546 (0.0784) | −0.0169 (0.00383) | 2.39E-09 | TRIM39-RPP21 | No | No |
| 2 | 240033164 | 240033412 | 249 | 3 | - - - | 0.913 (0.0262) | −0.0108 (0.00740) | 6.71E-04 | HDAC4 | No | No |
| 21 | 32649155 | 32649215 | 61 | 3 | - - + | 0.946 (0.00710) | −0.00124 (0.00333) | 0.005991 | TIAM1 | No | No |
| 19 | 2250901 | 2251067 | 167 | 4 | - - - - | 0.142 (0.0837) | −0.00592 (0.00251) | 0.007136 | AMH | No | Yes |
| 1 | 215804951 | 215805150 | 200 | 2 | - - | 0.725 (0.156) | −0.0121 (0.00434) | 0.007565 | USH2A | No | No |
| 16 | 85007113 | 85007184 | 72 | 2 | - - | 0.892 (0.0622) | −0.00729 (0.00539) | 0.01132 | ZDHHC7 | No | No |
| 11 | 464353 | 464554 | 202 | 2 | - - | 0.949 (0.0223) | −0.00539 (0.000677) | 0.01232 | PTDSS2 | No | No |
| 9 | 2717325 | 2717531 | 207 | 3 | + - - | 0.952 (0.00815) | −0.00357 (0.00303) | 0.0153 | KCNV2 | Yes | No |
| 7 | 36500566 | 36500624 | 59 | 2 | + - | 0.948 (0.00193) | −0.000239 (0.00394) | 0.02489 | ANLN | No | No |
| 14 | 65231319 | 65231406 | 88 | 2 | − | 0.500 (0.177) | −0.0101 (0.00722) | 0.02852 | SPTB | No | No |

Differentially methylated regions were identified using comb-p, which assesses the spatial correlation of P values obtained from the conditional regression model. Šidák-corrected P value accounts for multiple comparisons using the comb-p algorithm. Regions encompassing at least two probes with Šidák-corrected P < 0.05 were considered significant. The coefficient direction shows positive and negative coefficients for probes within the region based on genomic position. Mean beta and delta-beta values for probes within each region are shown. A region associated with *AMH* overlaps with a CoRSIV region and has been previously identified to harbor abnormal DNA methylation at diagnosis. *Chr* Chromosome. *CoRSIV* Correlated region of systemic interindividual variation. *SD* Standard deviation.

across the full set of array probes (57.7% negative coefficients). To further evaluate functional elements associated with CoRSIV probes, we accessed the locations of 852,830 candidate cis-regulatory elements (cCRE) from the UCSC Genome Browser. 88,223 probes from the full array overlapped with 63,130 distinct cCRE regions, while 670 CoRSIV probes overlapped with 315 cCRE regions. The coefficient direction was uniform in the full array, with a significant negative bias in all regulatory element classes. Negative coefficients made up 69–70.2% of the total probes across all cCRE classes. In contrast, in the CoRSIV-associated probes, a significant negative bias was noted in distal enhancer sites only, while proximal enhancer and promoter sites were significantly biased in a positive direction (Supplementary Fig. 9b).

## Discussion

This is the first epigenome-wide investigation of DNA methylation at birth in discordant monozygotic twins and the risk of pediatric ALL. We identified a total of 240 significant DMPs and 10 DMRs associated with the development of pediatric ALL in our twin data set, including 20 DMPs and one DMR overlapping with genes known to have aberrant DNA methylation in ALL at diagnosis[14]. We further confirmed these findings in a sample of four significant DMPs using DNA methylation-specific ddPCR, indicating these results are unlikely to result from experimental artifacts. The identification of these significant probes and regions overlapping epigenetically dysregulated genes in ALL at diagnosis supports a potential early role of aberrant DNA methylation in leukemic transformation. Furthermore, these significant sites were identified across the diverse group of ALL diagnoses evaluated in this study, a condition with no relationship itself to ALL risk, further supporting a core, fundamental role in ALL development.

While no obvious functional associations were identified in gene set enrichment analysis of the significant DMPs and DMRs from our regression analysis, the within-pair analysis identified enrichment in immune-related terms. The top DMR identified in this study is located in the gene body of *TRIM39-RPP21*, a read-through transcript enjoining the N-terminus RING finger and B-box domains of *TRIM39* and *RPP21* and located within the major histocompatibility complex class I region of chromosome 6[22,23]. The nine associated probes in this region are hypomethylated in cases compared to controls. A member of the tripartite motif-containing (TRIM) family of proteins, TRIM39 negatively regulates NFKB signaling through stabilization of CACTIN in response to inflammatory stimulation through TNFα[24]. TRIM39 has been shown

to have additional roles in the regulation of cell cycle progression through interaction with p21[25], and to inhibit apoptosis through negative regulation of p53[26]. Variants in *TRIM39* are associated with chronic inflammatory and autoimmune disorders[27,28], including an association between hypomethylation around the promoter region of *TRIM39-RPP21* and inflammatory bowel disease[29].

We identified a strong pattern of global DNA hypomethylation in ALL cases compared to their unaffected twin siblings. Of the 41 twin pairs assessed, 30 demonstrated global hypomethylation in ALL cases compared to controls. These 30 did not differ significantly from the remaining 11 twin pairs in cases regarding sex, diagnosis code, age of leukemia onset, birthweight, or array batch/chip number. DNA hypomethylation was enhanced based on CpG context and genomic position, with stronger hypomethylation across open sea regions in comparison to CpG islands and promoter regions. These findings, with a global decrease in DNA methylation content and promoter-specific hypermethylation, align with the canonical view of epigenetic dysregulation in malignant cells[30], and are thought to induce chromosomal instability through de-repression of repetitive elements and inhibition of tumor suppressor genes[31,32]. While most studies evaluating global DNA hypomethylation in cancer focus on solid malignancies[31], this phenomenon is also identified in childhood ALL, however, inconsistently[17,33–36]. A recent analysis of the global DNA methylome of ALL demonstrates a lack of decreased DNA methylation content in T-ALL, while Ph-like, *DUX4*-rearranged, hypodiploid, and a group of unknown subtype B-ALL cases demonstrate subtle but consistent global hypomethylation in comparison to healthy precursor B and T-cells[37]. While the magnitude of global DNA hypomethylation in B-ALL appears far lower than in solid tumors reported in the study, the more moderate reported findings are similar to the results of the current investigation. Loss of DNA methylation in backbone regions of the epigenome, which largely overlaps with open sea regions, has been identified in ALL cells in comparison to B-cell progenitors and is particularly prevalent in the hyperdiploid subtype[17]. An evaluation of a single pair of monozygotic twins concordant for infantile *TCF3-ZNF384* translocated B-ALL identified similar patterns of global DNA hypomethylation across both twin pairs using whole genome sequencing[35]. Here, when evaluating discordant monozygotic twins, we see a similar DNA hypomethylation profile associated with ALL development, along with evidence of an early epigenetic divergence between those twins going on to develop ALL and those who do not. This feature is independent of the age of ALL development in our data set, indicating that

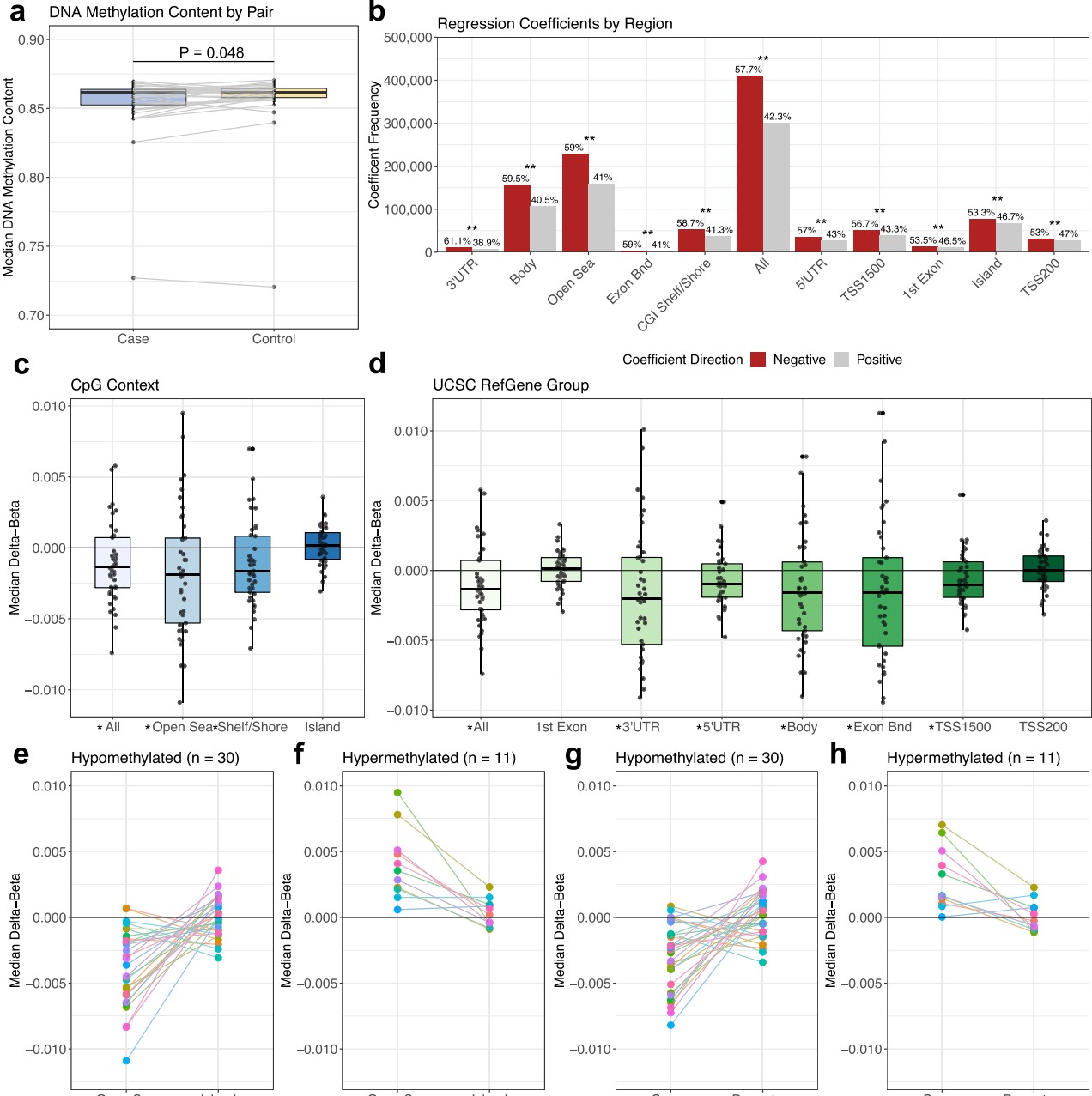

**Fig. 2 | Bias in negative coefficients from regression analysis indicates regionally specific DNA hypomethylation profile in ALL cases. a** Box and whisker plots demonstrate median DNA methylation content in $n = 41$ case and $n = 41$ control twins (pairs connected by gray lines). Significantly lower global DNA methylation was identified in the case of twins when compared to their sibling controls (Two-sided paired Wilcoxon test $P = 0.048$). **b** Bar plot demonstrating proportion of positive (indicating hypermethylation in cases) and negative (indicating hypomethylation) regression coefficients in all probes, by CpG island context, and by genetic context. The percentage of positive and negative probes are shown over bars. All regions are significantly biased toward negative regression coefficients. Regions are arranged by the strength of negative coefficient bias, with CpGs in 3′-UTR, gene body, open sea, exon boundaries (Exon Bnd), and CpG island shelf/shore regions showing a stronger negative bias than the overall array results. ** indicates FDR-corrected $P < 0.001$. Evaluation of DNA hypomethylation signature identified in conditional regression analysis using raw beta values paired by twin relationship (delta beta, or ALL-case DNA methylation beta value minus control beta value) according to **c** CpG context and **d** UCSC RefGene Group for $n = 41$ independent twin pairs. *FDR-corrected $P < 0.05$. The distribution of median delta beta values (case

beta minus control beta) for probes associated with each genomic region is shown for the full set of 41 twin pairs. A significant shift toward DNA hypomethylation in cases (negative delta beta values) was identified for probes associated with the overall array (All), as well as open sea and shelf/shore regions, however not in island regions. When looking at pair-specific median values (**e**), twin pairs with negative median delta beta values across all probes (globally hypomethylated in cases, $n = 30$) tended to have median island values near zero, demonstrating regional specificity of the identified DNA hypomethylation profile. Plot colors represent individual twin pairs. Similarly, by RefGene group (**f**), a significant negative shift was noted in the gene body, 3′-UTR, and 5′-UTR, at exon boundaries, and at TSS1500 sites, however, there was no negative shift associated with 1st exon and TSS200 probes. Among globally hypomethylated and hypermethylated cases, a pair-specific plot (**g**, **h**) shows specificity of findings between gene body and promoter-associated probes. Source data are provided as a Source data file. In boxplots, the box represents the interquartile range (IQR, first through the third quartile) with the centerline showing the median value for all subjects/twin pairs, whiskers show a minimum (first quartile minus 1.5 × IQR) and maximum (third quartile plus 1.5 × IQR) data range.

**Table 4 | Coefficient directional bias based on associated CpG region**

| Region | Coefficients | | | | Delta-Beta Values | | |
|---|---|---|---|---|---|---|---|
| | Negative (%) | Positive | Total | P value* | Median | IQR | P value** |
| Full array | | | | | | | |
| All probes | 409819 (57.7%) | 300191 | 710010 | 2.5E-323 | −0.00137 | 0.00353 | 0.00889 |
| CpG context | | | | | | | |
| Open sea | 228222 (58.9%) | 158886 | 387108 | 2E-323 | −0.00189 | 0.00599 | 0.0160 |
| Island | 76982 (53.2%) | 67578 | 144560 | 4.00E-135 | 1.66E-04 | 0.00187 | 0.555 |
| Shelf/shore | 52579 (58.7%) | 36971 | 89550 | 1E-323 | −0.00163 | 0.00396 | 0.00926 |
| UCSC RefGene group | | | | | | | |
| TSS200 | 30824 (53.0%) | 27326 | 58150 | 1.11E-47 | 1.56E-05 | 0.00183 | 0.6535 |
| TSS1500 | 50787 (56.7%) | 38788 | 89575 | 1E-323 | −1.00E-03 | 0.00250 | 0.0199 |
| 3'-UTR | 10793 (61.1%) | 6859 | 17652 | 3.44E-194 | −0.00201 | 0.00623 | 0.0207 |
| 5'-UTR | 35295 (57.0%) | 26589 | 61884 | 3.77E-269 | −9.53E-04 | 0.00239 | 0.0127 |
| Body | 156153 (59.5%) | 106283 | 262436 | 1.5E-323 | −0.00161 | 0.00492 | 0.0160 |
| Exon bnd | 2884 (59.0%) | 2006 | 4890 | 2.89E-36 | −0.00157 | 0.00635 | 0.0284 |
| 1st exon | 12511 (53.5%) | 10865 | 23376 | 5.11E-27 | 1.22E-04 | 0.00171 | 0.682 |
| Regulatory feature group | | | | | | | |
| Promoters | 27523 (52.0%) | 25416 | 52939 | 5.47E-20 | 2.57E-04 | 0.00230 | 0.456 |
| Genes | 1253 (60.1%) | 833 | 2086 | 3.42E-20 | −0.00134 | 0.00462 | 0.0143 |
| Non-genes | 381 (51.9%) | 353 | 734 | 0.319 | 1.90E-04 | 0.00187 | 0.868 |
| Unclassified | 22572 (56.2%) | 17589 | 40161 | 9.08E-137 | −4.02E-04 | 0.00168 | 0.226 |

Coefficient frequency is shown based upon probe associations with CpG island region (CGI Shelf/Shore, Island, Open Sea), UCSC RefGene Group (TSS1500, TSS200, 3'-UTR, 5'-UTR), and Regulatory Feature Group (Promoters, Genes) based on Illumina EPIC array probe annotations. Probes may overlap multiple regions. A significant negative bias in coefficients was identified in all groups by a two-sided binomial test*. While 57.7% of coefficients were negative across all probes, probes in CGI shelf/shore regions (58.7%), genes (60.1%), open sea (59.0%), and 3'-UTR (61.1%) were more highly biased toward negative coefficients compared to all probes on the array. In contrast, promoters (52.0%), TSS1500 (56.7%), TSS200 (53.0%), and island (53.3%) were less significantly biased. The median delta beta values of probes associated with the various regions are shown in the right half of the table. Delta-beta values were significantly shifted toward DNA hypomethylation (negative values) in cases in the overall array as well as specific regions including open sea probes (Wilcoxon signed-rank test**). In contrast, delta beta values were not significantly different from 0 in island, promoter, and TSS200 probes. *Exon Bnd* Exon boundary, *IQR* interquartile range, *TSS* transcriptional start site, *UTR* untranslated region.

decreased DNA methylation content may act as an early predisposing or priming step in leukemic transformation in some individuals. In contrast, other studies have identified a pattern of global DNA hypermethylation in childhood ALL[34] in comparison to peripheral blood from healthy control subjects, along with increased DNA methylation at LINE-1, Alu, and α-satellite elements[33,34]. Variable outcomes in these studies may be attributed to small sample sizes, molecular subtype-specific variations, or the use of inadequate controls, as lineage commitment in pre-B cells has been shown to be accompanied by demethylation of non-island regions[38]. In this study, the use of whole blood obtained from monozygotic twins, along with control for nucleated cell proportions in regression analysis, provides a more direct comparison of DNA methylation status.

We further identified evidence of early DNA hypomethylation across repetitive elements in ALL cases, however, this finding appeared to be driven by the strong degree of generalized open sea hypomethylation, rather than specific to repetitive element regions. Repeat DNA sequences, constituting approximately 56% of all CpGs, are genetic relics of transposons with the capacity to mobilize and reinsert throughout the genome when transcribed[39]. When actively transcribed, these elements may contribute to global chromosomal instability and oncogenesis[40]. LINE-1 elements make up the largest quantity of REs in the human genome, accounting for 500,000 copies, however, only 80–100 of these copies are likely to be transcriptionally active in an individual[41]. DNA hypomethylation has been described across all repetitive element classes in numerous malignancies[31]. A comparison of leukemic cells to B-cell progenitors showed repeat families were generally demethylated in ALL, however, satellite, tRNA, rRNA, simple repeat, and low complexity families were preferential de novo DNA methylated[17]. The pattern of DNA hypomethylation by raw beta values for ALL cases in this study was concordant across 18 of 19 repetitive elements classes, with the only non-significant class being

tRNA sequences. While the generally poor array coverage of repetitive regions limits more specific conclusions from this study, this result indicates these sites represent intriguing targets for further investigation.

While the exact mechanism is not identified in this study, there are multiple explanations for DNA methylation variation between monozygotic twins at birth. For one, the availability of intrauterine blood supply may vary between twins[42]. While we saw no significant difference in birthweight between cases and controls, more subtle discrepancies in maternal micronutrient availability may exist. This includes folate, which helps regulate the early establishment of DNA methylation[9,43], and has additionally been implicated as a risk factor in ALL development[44,45]. Random or stochastic influence on DNA methylation establishment might additionally contribute to the significant variations identified. As candidate metastable epialleles, the DNA methylation status of CoRSIVs is established in early development and influenced by the periconceptional environment and maternal nutrient availability[20]. We see a pattern of DNA hypomethylation across CoRSIV sites consistent with that seen across the full array. In contrast, we see evidence of DNA methylation variability in CoRSIVs associated with regulatory elements, with a pattern of DNA hypomethylation in distal enhancers and hypermethylation in proximal enhancers and promoters. While the poor array probe coverage of these regions limits further conclusions in this study, these results provide evidence of a non-random distribution of DNA methylation in CoRSIVs by functional elements in our twin data set. Presumably, these sites represent DNA methylation patterns established in the post-cleavage embryo, and contrast with twin epigenetic supersimilarity identified in established metastable epialleles originating prior to embryo cleavage[46].

There are multiple strengths in the design of this study. The use of identical monozygotic twins allows for greater power in identifying

significant variation in DNA methylation associated with the development of leukemia which might be undetectable in a group of genetically dissimilar ALL cases and controls. Significant DMPs identified in conditional regression analysis demonstrated relatively minimal absolute variation within pairs, instead reaching statistical significance due to subtle but consistent directional shifts in DNA methylation between cases and controls. Just 2 of 240 significant DMPs identified in the conditional regression analysis were also significant in the within-pair analysis, which identifies sites with large absolute DNA methylation variation within twin pairs. Furthermore, the twin study design controls ancestry, sex, and other shared birth factors which would not be present in a non-twin study. However, given the limitations of the cancer registry data used in this study, specific ALL subtype information was not available outside of denotation of B or T-cell lineages. Given the presence of subtype-specific DNA methylation profiles in ALL at the time of diagnosis, we were unable to assess whether this specificity is mirrored in DNA methylation profiles prior to the onset of leukemia (i.e., at birth). However, the presence of a core set of DNA methylation sites with aberrant DNA methylation suggests a commonality to the earliest steps in leukemogenesis, which would be theoretically identifiable without knowledge of the subtype of each included case in this study. In addition, coverage by the Illumina array of open sea regions, and in particular repetitive elements, is generally poor. This includes CoRSIVs, of which we identified overlaps with just 1128 of the 710,010 arrays CpGs. These results do, however, indicate a dramatic global depiction of DNA hypomethylation in ALL cases compared to their sibling controls, and based upon available data from the EPIC array we see consistency across these elements. The pan-hypomethylation signature identified across repetitive element classes implies the initiating demethylation process occurs concordantly within these regions, however further study is necessary to evaluate for specific variation in DNA methylation across repetitive element sequences.

In summary, we identified epigenetic variation in monozygotic twins which associates with the future development of ALL in one child but not their identical twin. While we identified a number of candidate DMPs and DMRs associated with ALL, the most striking is the profound degree of global and, specifically, open sea DNA hypomethylation identified in future ALL cases. These results call for further investigation of the potential variations occurring in twin gestations which might impart these notable epigenetic changes between otherwise genetically identical individuals. Given the unique paired design of this twin study, these findings may be subtle or indistinguishable in genetically dissimilar individuals. However, given the process of leukemic development in twins should be identical to that of singleton births, these results should be generalizable to all cases of pediatric ALL, and calls for further investigation of the role of early DNA methylation variation in ALL pathogenesis.

## Methods

### Study subjects
This study was approved by Institutional Review Boards at the California Health and Human Services Agency and the University of Southern California. Discordant twin cases of pediatric ALL were identified using linked records from the California Cancer Registry (CCR) and California Birth Statistical Master File (BSMF) spanning 1989 to 2015 based on reported ICD codes. Discordancy was defined by the identification of a singular case of ALL occurring within a twin pair, with the other sibling remaining unaffected to the end of the study period in 2015. Birth records were obtained for the case's twin and unaffected sibling. 148 twins discordant for ALL were identified in the combined registry data over this period. To obtain genomic DNA samples from subjects prior to the onset of ALL, archived neonatal blood spots (ANBS) were requested from the California Biobank for same-sex twin pairs. A total of 104 same-sex discordant ALL twin pairs

were identified in the linked CCR and BSMF registries, of which 86 had available ANBS samples for use in this study.

### Sample preparation and zygosity determination
Genomic DNA was extracted from two 4.7 mm card punches of each ANBS using the Beckman GenFind V3 kit and Eppendorf EpMotion 5075 (Eppendorf AG, HH, Germany). DNA concentrations ranged from 1.41–9.86 ng/μL (median 6.62 ng/μL) and ranged in volume from 32 to 50 μL (median 40 μL). Samples were subsequently randomized and submitted to ThermoFischer Scientific for analysis using the Axiom Precision Medicine Diversity Array (PMDA) genome-wide single-nucleotide polymorphism (SNP) array (ThermoFischer, Waltham, MA, USA). Zygosity status was subsequently assessed using an identity-by-descent analysis in PLINK (version 1.90) based on PMDA array, with 43 twin pairs confirmed to be monozygotic with pi-hat values ranging 0.9941–0.9998. The remaining 43 twin pairs were determined to be dizygotic, with pi-hat values ranging 0.4238–0.6116, and removed from further analysis.

### DNA-methylation array analysis
For the 43 identified monozygotic twin pairs, DNA samples were blocked randomized on 96-well plates and submitted to Diagenode, Inc. (Denville, NJ, USA) for bisulfite conversion using an in-house method (https://www.diagenode.com/en/categories/bisulfite-conversion) and for DNA methylation analysis using the Infinium Methylation EPIC genome-wide DNA-methylation array (Illumina, San Diego, CA, USA), with DNA volumes ranging 32.0–50 μL (median 40.0 μL) for total DNA amounts ranging 56.5–335.0 ng (median 272.0 ng). ALL cases and controls from individual twin pairs were randomly distributed on separate BeadChips (eight subjects per chip) for array analysis. Raw DNA methylation data files (IDAT) were imported into R (version 4.0.0, http://cran.r-project.org/). IDAT files were subsequently preprocessed and normalized using the openSesame pipeline from the "SeSAMe" package[47]. The distribution of signal background was calibrated on the Type I probe out-of-band signal. Probes with detection $P$ value >0.05 were masked from further analysis. NOOB background subtraction was performed, followed by removal of residual background and nonlinear scaling to correct for dye bias. Probes and subjects with more than 5% missing values were removed, with missing values imputed using the "impute.knn" function from the "impute" package[48]. Following normalization and data preprocessing, two twin pairs were observed to have significantly elevated detection $P$ values and were omitted from subsequent analysis. A total of 710,010 CpG probes passed quality control measures for inclusion in the analysis, including chromosomes X and Y. Zygosity status was confirmed using rs-labeled probes from the DNA methylation array for the 41 twin pairs per manufacturer-recommended protocols. tSNE analysis to evaluate data structure was conducted using package "Rtsne".

### Assessment of cell-type heterogeneity
Reference-based deconvolution of nucleated blood cell proportions was performed on all subjects using the Identifying Optimal DNA-methylation Libraries algorithm (IDOL)[49–51]. We used the "estimateCellCounts2" function in the "FlowSorted.Blood.EPIC" package in R and reference cord blood sample to estimate B-cell, CD4+ and CD8+ T-cell, monocyte, granulocyte, natural killer cells (NK), and nucleated red blood cells (nRBC) proportions in all subjects.

### Within-pair DNA-methylation assessment
To assess absolute differences in DNA methylation beta ($\beta$) values within individual twin pairs, we calculated the delta beta (case $\beta$ value minus control sibling $\beta$-value) for each probe on the array. We initially identified probes with absolute delta beta values greater than 0.15 across individual twin pairs. We subsequently identified recurrent probes (present in at least 2 twin pairs) across the entire group of

twins. We used the "gometh" function in the "missMethyl" package in R to evaluate for significantly enriched gene ontology and KEGG-pathway terms associated with recurrently DMPs from the within-pair analysis[52,53]. Within-pair analysis was additionally conducted on subsets of the full data set including B-cell lineage ($n = 30$ pair), unknown lineage ($n = 7$ pair), and T-cell lineage ($n = 4$ pair) ALL cases.

## Epigenome-wide association studies

To identify array probes associated with future development of ALL, we conducted a conditional regression analysis using the "survivor" package in R controlling for array plate and cell proportions estimated from deconvolution analysis to identify DMPs. Beta values were $\log_2$-transformed to M-values for this analysis. We controlled for the paired nature of the data set by adding a clustering term to the regression equation. We assessed $n = 37$ twin pairs with B-cell or unknown lineage ALL (omitting $n = 4$ T-cell ALL pairs) to improve data resolution. Regression output was annotated using the "IlluminaHumanMethylationEPICanno.ilm10b4.hg19" package in R. Significant DMPs were defined as FDR-corrected $P < 0.05$. DMRs were identified based upon the spatial correlation of $P$ values from the regression output using the "comb-P"[21] programs in Python (version 3.7.6) based on Šidák $P < 0.05$, which are corrected for multiple comparisons. The direction of effect was assessed by cross-referencing probes within each region with conditional regression coefficient results. Gene set enrichment analysis was performed on significant DMPs and DMRs using the "gometh" and "goregion" functions of the "missMethyl" package to assess GO and KEGG-pathway enrichment[52,53].

## DNA methylation-specific ddPCR analysis

To validate DNA methylation results generated from the EPIC array, performed DNA methylation-specific ddPCR on four significant DMPs (cg17080697 at *TRIM39*, cg14562331 at *CMIP*, cg04976226 at *FOXK1*, and cg11744295 at *SDHC*) in twin pairs with sufficient remaining genomic DNA sample availability ($n = 9$ pair). DMP targets were selected based upon significance in conditional regression assessment, low interclass correlation coefficients to maximize detectable differences between case and control twin, and the ability to generate suitable PCR primers and DNA methylation-specific probes for the ddPCR assay. Details of the DNA methylation-specific ddPCR assay are outlined elsewhere[54]. Briefly, DNA was reisolated from DBS for the 9 twin pairs and eluted to a total volume of 80 μL, with resultant DNA concentrations ranging 3.16–17.49 ng/μL by Picogreen. Samples were treated with sodium bisulfite using the EZ-96 DNA Methylation-Direct MagPrp Kit (Zymo Research Corporation, CA, USA) following preparation of the CT Conversion Reagent and into Section II of the manual's protocol (performed manually). Bisulfite-converted DNA was stored at −20 °C. To ensure DNA methylation-specific binding, PCR primers were designed for use with bisulfite-converted DNA using MethPrimer[54]. Distinct PrimeTime™ double-quenched probes were used to target methylated (5′ 6-FAM/ZEN/IBFQ 3′, FAM probe) and unmethylated (5′ HEX/ZEN/IBFQ 3′, HEX probe) DNA at the DMP site (Supplementary Data 18). Primers and probes were synthesized by Integrated DNA Technologies (IA, USA). Primers and probes were purified following standard procedures and resuspended in1x TE buffer (10 mM Tris, pH 8.0, 0.1 mM EDTA) to reach a total concentration of 100 μM and stored at −20 °C. All ddPCR reactions were performed using Bio-Rad's QX200 and AutoDG Droplet Digital PCR system (Bio-Rad Laboratories, CA, USA) according to the manufacturer's instructions. A total of 22 μl reactions were prepared with 5.5 μl Bio-Rad ddPCR 4X Multiplex Supermix, 1.1 μl of each 20× primer/probe mixture set (18 μM/5 μM; FAM and HEX), 1 μl of DNA at concentrations ranging 3.16–17.49 ng/μL and nuclease-free water in a 96-well plate. Droplet generation, amplification, and data acquisition followed Bio-Rad's rare event detection experimental guidelines with Channel 1 = FAM and Channel 2 = HEX. Thermal cycling conditions

followed Bio-Rad's ddPCR 4X Multiplex Supermix's procedures (for QX200) with an enzyme activation step of 10 min at 95 °C, followed by 40 cycles at 94 °C for 30 s and 60.0 °C for *TRIM39*, *CMIP* and *FOXK1*, or 52.1 °C for *SDHC* for 1 min, with an enzyme deactivation step for 10 min at 98 °C and an optional hold at 4 °C until use. The *SDHC* analysis was run in duplicate to ensure sufficient positive droplet count. Data analysis was performed using Bio-Rad's QuantaSoft Analysis Pro Software version 1.0596. Thresholds were manually set using the available automation tools.

## Assessment by genomic regions

We further assessed for bias in coefficient direction by genomic locations relationship to island regions, UCSC RefGene group, and Regulatory Feature Group as annotated by the Illumina Epic array annotation file. An exact binomial test was used to evaluate for significant bias in regression coefficients by negative and positive values. A similar analysis was conducted to evaluate for bias in coefficient direction based on overlaps with TF-binding motifs obtained from the ENCODE ChIP-seq database ($n = 149$ TF-binding motifs), and locations of repetitive element classes downloaded from the RepeatMasker library from the UCSC Genome Browser. To confirm bias in DNA methylation $\beta$ values associated with these results, we obtained median delta beta (case $\beta$ minus control $\beta$-values) for each associated region by individual twin pair and conducted a Wilcoxon signed-rank test to assess for a positive or negative bias in delta beta values (i.e., median delta beta values across all pairs significantly different from 0), by genomic region repetitive element or TF-binding motif. To determine whether TF-binding motifs were significantly enriched in our significant results from the conditional regression model compared to the full array, we used Fisher's exact test to compare the number of probes overlapping with individual TF motifs in significant CpGs and cross the full array. All statistical tests were corrected for multiple comparisons using a false discovery rate $P < 0.05$.

## Assessment of regulatory elements

To evaluate associations between our conditional regression results and CoRSIVs (correlated regions of systemic interindividual variation), we evaluated overlaps between the genomic locations of array probes and significant regions identified by comb-p with the locations of CoRSIVs[20]. To evaluate functional associations with these CoRSIV overlaps, we next assessed overlaps between these identified regions and 852,830 cCREs downloaded from the UCSC Genome Browser. We subsequently evaluated for a bias in the coefficient direction in probes associated with cCRE locations across the full array and in probes associated with both cCREs and CoRSIVs using an exact binomial test.

## Reporting summary

Further information on research design is available in the Nature Research Reporting Summary linked to this article.

# Data availability

This study used biospecimens from the California Biobank Program. Per California Health and Safety Code Sections 124980 (j), 124991 (b), (g), (h), and 103850 (a) and (d), which protect the confidentiality of data obtained from biospecimens, we are respectfully unable to share raw, individual-level genomic and genome-wide DNA methylation data reported in this study, which are property of the State of California. Should we be contacted regarding individual-level data contributing to the findings reported in this study, inquiries will be directed to the California Department of Public Health Institutional Review Board to establish an approved protocol to utilize the data, which cannot otherwise be shared peer-to-peer. The State of California has provided guidance on data sharing per the following statement: "California has determined that researchers requesting the use of California Biobank

biospecimens for their studies will need to seek an exemption from NIH or other granting or funder requirements regarding the uploading of study results into an external bank or repository (including into the NIH dbGaP or other bank or repository). This applies to any uploading of genomic data and/or sharing of these biospecimens or individual data derived from these biospecimens. Such activities have been determined to violate the statutory scheme of California Health and Safety Code Section 124980 (j), 124991 (b), (g), (h), and 103850 (a) and (d), which protect the confidential nature of biospecimens and individual data derived from biospecimens. All investigators seeking to use California specimens for projects or grant-related activities that require or seek such sharing (at the NIH or elsewhere) must seek an exemption from genomic data sharing requirements. If such an exemption is not secured, samples will not be released to an investigator. Investigators may agree to share aggregate data on SNP frequency and their associated *P*-values with other investigators and may upload such frequencies into repositories including the NIH dbGaP repository providing (a) the denominator from which the data is derived includes no fewer than 20,000 individuals; (b) no cell count is for <5 individuals; and (c) no correlations or linkage probabilities between SNPs are provided." Datasets evaluated in this manuscript include the California Cancer Registry and California Birth Records Master Statistical File. Source data for Figs. 1 and 2, and Supplementary Figures 1 and 2 are provided as a Source data file with this paper. Conditional logistic regression results are presented in Source data file (under Fig. 1a) and in annotated form as Supplementary Data 5. The delta beta values (leukemia case DNA methylation beta value minus control beta value) generated in this study have been deposited in the Harvard Dataverse (https://doi.org/10.7910/DVN/NT1WAX)[55]. Source data are provided with this paper.

## Code availability

Code used in the conditional logistic regression analysis is publicly available for download from the Harvard Dataverse (https://doi.org/10.7910/DVN/NT1WAX)[55]. The remaining code used for this analysis is made available on request to the corresponding author.

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

## Acknowledgements

We would like to thank Robert Waterland, Ph.D., professor of Pediatrics-Nutrition at Baylor College of Medicine for his contribution in providing the regional locations of CoRSIV sites utilized in this study and for scientific discussions and advice. We would like to thank Wendy Cozen, DO for additional discussions and advice. We thank Robin Cooley, MSc, for expert assistance in collection of neonatal blood spots. The collection of cancer incidence data used in this study was supported by the California Department of Public Health as part of the statewide cancer reporting program mandated by California Health and Safety Code Section 103885; the National Cancer Institute's Surveillance, Epidemiology and End Results Program under contract HHSN261201000140C awarded to the Cancer Prevention Institute of California, contract HHSN261201000035C awarded to the University of Southern California and contract HHSN261201000034C awarded to the Public Health Institute; and the Centers for Disease Control and Prevention's National Program of Cancer Registries, under agreement U58DP003862-01 awarded to the California Department of Public Health. The ideas and opinions expressed herein are those of the author(s) and endorsement by the State of California, Department of Public Health, the National Cancer Institute, and the Centers for Disease Control and Prevention or their Contractors and Subcontractors is not intended nor should be inferred. We thank the Hyundai Hope on Wheels foundation for their support of author EMN. We additionally thank The Saban Research Institute Research Career Development Fellowship and The George Donnell Society for Pediatric Scientists at Children's Hospital Los Angeles for their support. We thank the V Foundation for support in the initial linkage for twin finding (Grant FP067172) awarded to author J.L.W.

## Author contributions

E.M.N. and J.L.W. designed the experiment; J.L.W., E.M.N., and A.J.D. oversaw the experiment and data interpretation; Q.F. and E.M.N. abstracted registry data; E.M.N., S.S.M., K.A. conducted the experiments; E.M.N. and S.L. conducted data analysis; K.S. contributed to the statistical design of the study; E.M.N. wrote the manuscript with input and approval of all authors.

## Competing interests

The authors declare no competing interests.

## Ethics

The procedures for obtaining archived dried blood spots (DBS) from the California Department of Public Health (CDPH) have already passed human subjects approval at the State of California institutional review board (IRB) with reliances at other institutions, including IRB approval from the University of Southern California (USC). The CDPH Genetic Diseases Screening Branch obtains DBS from all neonates born within the state with the purpose of the Newborn Screening Program (NBS). The following use in the NBS, the remaining samples are archived under the California Biobank Program and are made available for use in appropriate scientific investigations under an "opt-out" mechanism per Section 6505 of Title 17 of the California Administrative Code. Subjects and parents of subjects are thus not required to opt-in to selected appropriate scientific investigations. Samples are stored and can be accessed for research purposes with approval from the California Health and Human Services Agency's (CHHSA) Committee on the Protection of Human Subjects (CPHS). Approval from the CPHS for samples to be used in this proposal has been obtained under protocol #17-04-2958.
