## [Peer Review File · Nature Communications]

DNA Methylation at Birth in Monozygotic Twins Discordant for Pediatric Acute Lymphoblastic LeukemiaREVIEWER COMMENTS

Reviewer #1 (Remarks to the Author):

The authors conducted an epigenome-wide association study of neonatal blood spot DNA from identical twins, only one of each pair subsequently developing acute lymphoblastic leukaemia. They identified significant CpGs and regions associated with a future diagnosis of ALL.

In general the manuscript this was easy to understand, with a clear introduction and rational, valid and justified methods results that flowed in a logical manner and a thorough discussion. However, there were some exceptions, which I outline below.

1. General comment: as the study was conducted on samples collected soon after birth, sex would have been known but not gender will. Can the authors please change all mentions of 'gender' to 'sex'?
2. General comment: throughout the manuscript, the authors mention CoRSIVs and/or metastable epialleles. On their first mention, can they clarify whether they are the same thing or different?
3. Please add tissues studied and the methylation analysis platform to the abstract
4. Line 65: please change 'heritable' to 'mitotically heritable'
5. Lines 144-157: I have doubts about the validity of this section and would lie the authors to justify and it and reference to such an approach For example in lines 147-8 the authors say, "Probes meeting a threshold absolute delta beta value difference of 0.15 or greater were considered significantly different" . This looks like ranking by effect size, which does not involve using a statistical test, so why 'significant'? Also, why didn't the authors compare their effects-size 'within-pair assessment' data with their regression analysis data?

Tables 2 and 3 should include delta betas & the magnitudes should be discussed in the text and compared to those seen in diagnostic samples in other studies or the same patients.

Lines 168-9: "Of these significant probes, 3 overlapped with probes identified as harbouring constitutional differential DNA methylation in ALL at diagnosis ". Is this in general or in this cohort?

Fig 1 mentions Q-Q plots and inflation. These results should be presented in the Results section.

Lines 377-378 "intrauterine blood supply may vary between twins" needs a citation

In the discussion the authors should acknowledge that stochastic factors could be at plan in addition/instead of twin environment-specific factors.

Reviewer #2 (Remarks to the Author):

In this paper, the authors analyze the DNA methylation levels of whole blood (via Guthrie cards) at birth of 41 monozygotic twin pairs discordant for ALL later in life. This is a very rare and exciting sample set, making this the largest study of DNA methylation analysis on discordant ALL monozygotic twin pairs to date. The hypomethylation profile is intriguing, but there are several technical and methodological clarifications needed to strengthen and support the conclusions of this study:

1. The data presented in this manuscript uses the robust EPIC array. However, as noted by the authors the EPIC arrays only covers ~3% of the CpG sites in the human genome. Given the relatively small sample size of the cohort and 1:1 paired case:control study design, it isn't well motivated why a true genome-wide approach (i.e. WGBS) wasn't implemented.

2. Not enough detail is provided about how the DNA methylation data were derived and there are some blatant errors that must be corrected. For example, please explain how was the genomic DNA quantified (nanodrop, qubit, ?). How many nanograms were used in the bisulfite conversion, what method was used for bisulfite conversion, and was bisulfite conversion done in one or several batches (in 96-well plate or tubes?). How many nanograms were subsequently loaded on the EPIC arrays? Can the "block randomization" method used for the arrays be clarified/outlined? On line 465 it is stated that samples were sent for "bisulfite sequencing" and methylation analysis. As far as I can tell no bisulfite sequencing data was presented in the manuscript. The actual empirical cut-offs for sample

and probe filtering must be given, i.e. for the “significantly elevated detection P values” on line 469 and for the “poor performing microarray probes” on line 471. How were probes filtered/masked if the detection p-value was non-significant and what was the cut-off? Were X and Y chromosomes included in the analysis? Importantly, was probe- and/or sample normalization performed? If not, why? No QC plots are shown, for example a PCA of the entire dataset showing the relationship between the 41 (or 43 to show the outliers) twin pairs to one another should provide at minimum to show that there are not major batch effects in this dataset.

3. Although the authors address the reason behind why the data are not openly available, which is understandable, they should aim to provide as much raw/ or beta-value level data as possible. I do not see the reason why only highly processed data are shown, for example median of delta-beta values, but no actual individual methylation levels/beta-values. The median delta beta-values do not say anything about if the change was from in regions of high methylation levels or low- and how the methylation levels of these regions compared amongst the individuals. This leads the reader to wonder if the methylation changes in regions of methylation variability in this cohort or in regions with little inter-individual methylation difference? If actual beta-values could be plotted/shown for at least the top DMPs, for example, this could improve confidence in the results presented. Median/mean beta-values and range could be shown in the tables, ie table 2/3?, and sample-level beta-values should be able to be presented in the supplemental tables for the most significant CpG sites- which shouldn't be affected by underlying genetic differences and thus would not be able to be used to identify an individual.

4. In Figure 2d/f, why do the lines cross each other so consequently for the “hypomethylated cases”? The twin pairs with lowest median delta beta-value in open sea tend to have amongst the highest delta beta-values in island, while those with the highest in open sea have the lowest in island. Why is this? Is this true biology or some sort of batch effect or something to do with CpG density on the arrays?

5. Case characteristics/Table 1: There appears to be a major bias towards female twin pairs. Is it a sample bias or are cases more prevalent amongst female monozygotic twins? What does the asterisk mean by the p-value next to birthweight? Is it possible to provide more granular level of information about the ALL diagnoses? Not otherwise specified (NOS) is not explained nor is it clear why only one patient is indicated to have t(12;21)ETV6-RUNX1 positive ALL, when approximately 20% of cases should have this aberration?

6. A general complaint about the manuscript is the extensive focus on gene enrichment and genomic location analyses. These analyses are indeed valuable, but are too lengthy in the main text/figures and the paper could be made sharper if much of those analyses were moved to the supplement and only the main take home points would be presented in the main text.

7. The figures can be generally improved to be more informative and easier to read. Specifically, for Figure 2: the plots are too large (with excessive white space) and text size on the axes too small, making them hard to read. Some of the results would be better suited in a table (panels A & B for example). The box plots (C & E) would be improved with actual median data points. The figure legend says panel C contains the distribution of median delta beta-values but the box plot doesn't do a good job showing the distribution of these data points. Perhaps the actual distribution of delta beta values (not the median) with a violin plot or similar would do a better job depicting this. Panels D and F are lacking explanation of the color scheme, which seems a bit random. The large overlap in data points could be made clearer by adding jitter to separate the overlapping points.

8. How/why were a total of 37 twin pairs included in the conditional regression analysis for B-ALL when in table 1 it is stated that 32 cases are B-cell, 4 are T-cell and the remaining 7 were not listed or unknown? The decision to remove the four cases who later develop T-ALL is perhaps a little confusing given that it seems that the some of the cases with unknown cell lineage were included in the

regression analysis anyways.

9. The manuscript is lacking a discussion/analysis of time to ALL diagnosis and magnitude of global DNA hypomethylation. On lines 251-253, it was noted that the 30 twin pairs that demonstrated a "universal hypomethylation" didn't differ in terms of some phenotypes, but it was not mentioned if they differed in time to ALL diagnosis, which would have been interesting to look at. Importantly, processing batch of bisulfite conversion/location on chip or other technical reason for lower DNA methylation level should be looked at.

10. It is further unclear why a simple "global" assessment of DNA methylation % was not calculated and compared within a pair and also between individuals/twin pairs? Twins are assumed to show more similar DNA methylation to one another than in comparison to unrelated individuals, is this the case in this dataset?

11. Given the rarity and difficulty in obtaining these sample types, validation on external samples is understandably difficult. However, some level of technical validation of some of the findings would be required at a minimum, to make sure that these observations are technically reproducible and not due to an unaddressed batch effect. Especially given the generally small DNA methylation differences within the twins observed.

12. The references in the introduction could use updating. Many are 10-20 years old and there are many more up to date references that would be suitable for describing the current situation in the ALL field.

13. How do the authors see their results in comparison to other similar studies in other pediatric hematological malignancies (ie. Behnert et al, 2021 Leukemia, JMML, for example).

May 6, 2022

Response to Reviewer Comments

Dear Reviewers,

We thank the reviewers for their positive comments, and below we provide detailed responses to their specific concerns that we believe have strengthened our manuscript, "DNA Methylation at Birth in Monozygotic Twins Discordant for Pediatric Acute Lymphoblastic Leukemia." The revised manuscript includes a new analysis, DNA methylation-specific droplet digital PCR, conducted with the focus of validating the results obtained from the EPIC DNA methylation array. Specific discussion of this analysis is outlined below and within the revised manuscript, where we describe significant concordance between the new analysis and EPIC array results. We feel this analysis further supports the biological relevance of reported findings. Furthermore, in response to the reviewer comments we sought to provide further details of the datasets assessed in the manuscript. We added median DNA methylation values to tables and have provided a full report of the results obtained from the conditional regression analysis described in the manuscript as a supplementary table. We have also added a depiction of individual-level data for subjects in the study in the supplementary files. Finally, in response to these comments we sought to improve clarity in the description of our methodologies and results, including moving two main body figures to the supplementary files to better focus the manuscript to the main conclusions described herein.

Reviewer #1 (Remarks to the Author):

The authors conducted an epigenome-wide association study of neonatal blood spot DNA from identical twins, only one of each pair subsequently developing acute lymphoblastic leukaemia. They identified significant CpGs and regions associated with a future diagnosis of ALL.

In general the manuscript this was easy to understand, with a clear introduction and rational, valid and justified methods results that flowed in a logical manner and a thorough discussion. However, there were some exceptions, which I outline below.

1. General comment: as the study was conducted on samples collected soon after birth, sex would have been known but not gender will. Can the authors please change all mentions of 'gender' to sex'?

We thank the reviewer for this suggestion, and all mentions of 'gender' in the paper have now been changed to 'sex.' Please see lines 360, 435, 494, 501 and Table 1.

2. General comment: throughout the manuscript, the authors mention CoRSIVs and/or metastable epialleles. On their first mention, can they clarify whether they are the same thing or different?

Clarification regarding this issue has now been made in the introduction of the manuscript. In brief, metastable epialleles are defined as sites of epigenetic variation established stochastically early in embryogenesis and maintained systemically throughout subsequent cellular differentiation. These sites are established stochastically with sensitivity to environmental influence, but largely without genetic influence, thus acting as epigenetic polymorphisms. CoRSIVs were initially defined purely as sites of DNA methylation correlated across tissues which show interindividual variation. While also established stochastically in early embryogenesis and maintained systemically throughout cellular differentiation, these sites may be influenced by a combination of environmental and potentially genetic effects. Thus, a subset

of CoRSIVs may act as true metastable epialleles; however this has not been definitively demonstrated to date.

To better provide clarity on this issue for readers, a revision to the introduction at lines 94-101 has been made.

3. *Please add tissues studied and the methylation analysis platform to the abstract*

Information regarding these two items has been added to the abstract at line 29 in the abstract.

4. *Line 65: please change 'heritable' to 'mitotically heritable'*

Line 65 has been changed to reflect 'mitotically heritable.'

5. *Lines 144-157: I have doubts about the validity of this section and would like the authors to justify and it and reference to such an approach For example in lines 147-8 the authors say, "Probes meeting a threshold absolute delta beta value difference of 0.15 or greater were considered significantly different" . This looks like ranking by effect size, which does not involve using a statistical test, so why 'significant'? Also, why didn't the authors compare their effects-size 'within-pair assessment' data with their regression analysis data? (Author note – Lines 144-157 are now 155-168 in the revised manuscript).*

For the within-pair analysis, we set a threshold of 0.15 for CpG-specific DNA methylation difference between case and control twins for inclusion in analysis as sufficiently variable. The word 'significant' has been removed from this section (line 158) to reflect that a statistical test was not used as a mean to identify these sites. There is no universally accepted threshold for within-pair beta value variation using the EPIC or 450k array. Previous studies conducting a similar "within pair" design have defined delta-beta significance threshold varying from 0.15 to 0.5 (please see below references 1-3).

Mean delta beta values have been added to the conditional regression analysis data as part of the supplementary data files (Specifically, Supplementary Tables 2 and 5). We conducted the two separate analyses not to demonstrate consistency between the two approaches, but rather to highlight pair-specific sites of strong variation (through within-pair analysis) and consistent direction of variation using the conditional regression approach. A comparison between conditional regression coefficients and probes meeting the delta beta threshold of 0.15 in the within pair analysis has been added to Supplementary Table 5. Notably, only 2 of 240 significant probes from conditional regression analysis were also identified in the within pair analysis as having substantial intrapair beta differences. This further demonstrates the value of using separate analytical approaches, as the conditional regression analysis appears to better identify subtle but consistent directional shifts in methylation between cases and controls across the data set, while the within-pair analysis identified sites with large methylation shifts within a subset of twin pairs analyzed. We have further noted this in the discussion section of the paper at lines 429-434.

References:

1. Kaut O, Schmitt I, Tost J, Busato F, Liu Y, Hofmann P, Witt SH, Rietschel M, Fröhlich H, Wüllner U. Epigenome-wide DNA methylation analysis in siblings and monozygotic twins discordant for sporadic Parkinson's disease revealed different epigenetic patterns in peripheral

blood mononuclear cells. *Neurogenetics*. 2017 Jan;18(1):7-22. doi: 10.1007/s10048-016-0497-x. Epub 2016 Oct 6. PMID: 27709425.

2. Souren NY, Gerdes LA, Lutsik P, et al. DNA methylation signatures of monozygotic twins clinically discordant for multiple sclerosis. *Nat Commun*. 2019;10(1):2094. Published 2019 May 7. doi:10.1038/s41467-019-09984-3

3. Mohandas N, Bass-Stringer S, Maksimovic J, et al. Epigenome-wide analysis in newborn blood spots from monozygotic twins discordant for cerebral palsy reveals consistent regional differences in DNA methylation. *Clin Epigenetics*. 2018;10:25. Published 2018 Feb 23. doi:10.1186/s13148-018-0457-4

Tables 2 and 3 should include delta betas & the magnitudes should be discussed in the text and compared to those seen in diagnostic samples in other studies or the same patients.

Delta beta values have been added to Table 2 and 3. Supplementary Table 8 has been added to demonstrate mean beta and delta beta values for each probe within significant differentially methylated regions for reference.

The magnitude of delta beta values identified in conditional regression analysis are small – generally indicating subtle shifts in DNA methylation rather than large magnitude variations between case and control twins. Consistent with the response to the previous question, this is in line with the intent of the conditional logistic regression approach, which allows for detection of slight but consistent directional variation in DNA methylation between cases and controls. For this reason, we find the consistent, significant shift toward DNA hypomethylation in case twins evidenced by the results of the conditional regression analysis to be notable.

To better reflect this for readers of the manuscript, a discussion on this matter has been added to lines 429-434 within the discussion section.

Lines 168-9: “Of these significant probes, 3 overlapped with probes identified as harbouring constitutional differential DNA methylation in ALL at diagnosis “. Is this in general or in this cohort? (Note – Lines 168-169 now refers to line 185-186 in the revised manuscript).

The significant probes identified in this study were located within genes previously described (in reference 14) to harbour constitutional (across all ALL subtypes assessed in the prior study) differential DNA methylation at diagnosis in ALL blasts. As we did not have access to diagnostic leukemia samples for this cohort, we were not able to make a similar comparison specific to subjects included in this study.

Fig 1 mentions Q-Q plots and inflation. These results should be presented in the Results section.

Specific reference to the Q-Q plot and genomic inflation have now been added to the main results section. Please refer to line 182.

Lines 377-378 “intrauterine blood supply may vary between twins” needs a citation (Author note – this now refers to lines 407-408).

The below noted citation has been added to this section as reference 41.

Reference:

Salafia, C. M. & Maas, E. The twin placenta: framework for gross analysis in fetal origins of adult disease initiatives. *Paediatr Perinat Epidemiol* 19 Suppl 1, 23-31, doi:10.1111/j.1365-3016.2005.00576.x (2005).

In the discussion the authors should acknowledge that stochastic factors could be at plan in addition/instead of twin environment-specific factors.

Acknowledgement of a role for stochastic factors influencing DNA methylation in twins has been made in the discussion. Please refer to lines 412-413.

Reviewer #2 (Remarks to the Author):

In this paper, the authors analyze the DNA methylation levels of whole blood (via Guthrie cards) at birth of 41 monozygotic twin pairs discordant for ALL later in life. This is a very rare and exciting sample set, making this the largest study of DNA methylation analysis on discordant ALL monozygotic twin pairs to date. The hypomethylation profile is intriguing, but there are several technical and methodological clarifications needed to strengthen and support the conclusions of this study:

1. The data presented in this manuscript uses the robust EPIC array. However, as noted by the authors the EPIC arrays only covers ~3% of the CpG sites in the human genome. Given the relatively small sample size of the cohort and 1:1 paired case:control study design, it isn't well motivated why a true genome-wide approach (i.e. WGBS) wasn't implemented.

We selected the EPIC array for this investigation due to the relatively limited amount of genomic DNA isolated from newborn dried blood spots for use in analysis and to best maximize the coverage of genes and functional elements of the human genome. While other approaches including whole genome bisulfite sequencing (WGBS) were considered, we identified several drawbacks which we felt would limit our analysis. For one, low-pass WGBS would preclude adjustment for nucleated cell proportions in the whole blood samples evaluated in this study (as noted in the reference listed below), while the expense of higher-depth WGBS was prohibitive due to the large number of samples in our study. Additionally, we felt coverage of the EPIC array was sufficient to identify significant DNA methylation variation in twins at genes and functional sites of interest in the study.

Furthermore, we feel the strength of our results from an array-based approach, including significant evidence of DNA hypomethylation in ALL cases, is well supported despite 3% coverage of CpGs in the human genome. As noted in our response to comment 11 below, we validated the results of our array-based approach using DNA methylation-specific droplet digital PCR to directly sequence a set of significant CpG sites. We further demonstrate consistency of the two methodologies in the manuscript, providing further evidence supporting for our initial results. Please see lines 205-223.

Reference:

Laufer BI, Hwang H, Jianu JM, et al. Low-pass whole genome bisulfite sequencing of neonatal dried blood spots identifies a role for RUNX1 in Down syndrome DNA methylation profiles. *Hum Mol Genet.* 2021;29(21):3465-3476. doi:10.1093/hmg/ddaa218

2. Not enough detail is provided about how the DNA methylation data were derived and there are some blatant errors that must be corrected. For example, please explain how was the genomic DNA quantified (nanodrop, qubit, ?). How many nanograms were used in the bisulfite conversion, what method was used for bisulfite conversion, and was bisulfite conversion done in one or several batches (in 96-well plate or tubes?). How many nanograms were subsequently loaded on the EPIC arrays? Can the “block randomization” method used for the arrays be clarified/outlined? On line 465 it is stated that samples were sent for “bisulfite sequencing” and methylation analysis. As far as I can tell no bisulfite sequencing data was presented in the manuscript. The actual empirical cut-offs for sample and probe filtering must be given, i.e. for the “significantly elevated detection P values” on line 469 and for the “poor performing microarray probes” on line 471. How were probes filtered/masked if the detection p-value was non-significant and what was the cut-off? Were X and Y chromosomes included in the analysis? Importantly, was probe- and/or sample normalization performed? If not, why? No QC plots are shown, for example a PCA of the entire dataset showing the relationship between the 41 (or 43 to show the outliers) twin pairs to one another should provide at minimum to show that there are not major batch effects in this dataset.

We thank the reviewer for highlighting this issue, and have attempted to address their concerns as follows.

The methods section has been updated to better describe the process of DNA isolation and application of the EPIC array, which was done by an outside facility using genomic DNA supplied by our laboratory. The outside facility additionally conducted bisulfite treatment of the DNA samples as part of their processing. Please see lines 518 for this change. For more specific information regarding the bisulfite treatment process used by the outside facility, please see the following reference which has now been added to the manuscript at lines 518-519: <https://www.diagenode.com/en/categories/bisulfite-conversion>. The mention of bisulfite sequencing has been removed from the methods section. The concentrations of genomic DNA isolated are referenced at lines 506-507, and the amount of DNA sent for bisulfite treatment and array analysis are now provided at line 521-522.

Block randomization is more specifically outlined in lines 517-518 and 522-524, noting that twin pairs (case and control) were randomly distributed on separate chips (8 subjects per chip) whereas the bisulfite treatment was performed by plate (96 samples; indeed, one bisulfite treatment for the whole sample set).

We now provide further description of the quality control and normalization process as follows: We employed a threshold detection-P value > 0.05 to filter poorly performing probes. We removed probes and samples with missingness $> 5\%$. Both X and Y chromosomes were included in the analysis, which is now referenced in the manuscript. This information has been further detailed in lines 525-534.

To better demonstrate DNA methylation relationships between twin pairs, we have provided additional supplementary figures (Supplementary Fig. 2-4) and noted strength of correlation between twin pairs in the manuscript at lines 146-152. We demonstrate consistent and high correlation in beta values across the full set of array probes within twin pairs in Supplementary Fig. 2 (R values ranging 0.968-0.991). We further demonstrate correlation remains high (R values ranging 0.900 to 0.990) even across the top 1000 most variable probes within twins

(Supplementary Fig. 3). As a comparison, we additionally demonstrate a similar analysis of 5 randomly matched individuals in the data set to represent beta correlation in unrelated individuals in Supplementary Fig. 4. Notably, while correlation across the entire array is high (R values ranging 0.964-0.984), this correlation was far reduced in the top 1000 most variable probes compared to twin pairs (R values ranging 0.596-0.675). These plots additionally serve to allow the reader to visually evaluate distribution of raw data by twin pair utilized in the analysis.

3. *Although the authors address the reason behind why the data are not openly available, which is understandable, they should aim to provide as much raw/ or beta-value level data as possible. I do not see the reason why only highly processed data are shown, for example median of delta-beta values, but no actual individual methylation levels/beta-values. The median delta beta-values do not say anything about if the change was from in regions of high methylation levels or low- and how the methylation levels of these regions compared amongst the individuals. This leads the reader to wonder if the methylation changes in regions of methylation variability in this cohort or in regions with little inter-individual methylation difference? If actual beta-values could be plotted/shown for at least the top DMPs, for example, this could improve confidence in the results presented. Median/mean beta-values and range could be shown in the tables, ie table 2/3?, and sample-level beta-values should be able to be presented in the supplemental tables for the most significant CpG sites- which shouldn't be affected by underlying genetic differences and thus would not be able to be used to identify an individual.*

To better provide the reader with raw data for evaluation considering our restrictions in the dissemination of individual-level data, we have attempted to make alterations reflecting the above suggestions. An additional supplementary figure has been added (Supplementary Fig. 5) to show a comparison of raw beta values in cases and controls by twin pairing for the top 20 DMPs identified in the regression analysis. This is noted in the manuscript in lines 180-181. We have added mean beta values to Tables 2 and 3, along with mean delta beta values to demonstrate DNA methylation variation by case status. We additionally edited Supplementary Table 5 to show mean beta values for all subjects, for cases only, for controls only, and mean delta-beta values for all probes assessed in the study. We have also provided mean beta and delta-beta values to Supplementary Tables 2 and 8. We have also added Supplementary Table 13 which lists median global DNA methylation content for case and control individuals in the study by twin pairing. We made these changes with the aim of providing further information for the reader regarding DNA methylation values and variation at these regions while maintaining confidentiality of individual-level data from the analysis.

4. *In Figure 2d/f, why do the lines cross each other so consequently for the “hypomethylated cases”? The twin pairs with lowest median delta beta-value in open sea tend to have amongst the highest delta beta-values in island, while those with the highest in open sea have the lowest in island. Why is this? Is this true biology or some sort of batch effect or something to do with CpG density on the arrays?*

We interpreted this to argue that this is less “batch effect” and more a true biological effect based on specificity of hypo versus hypermethylation by region. If batch effect from the array were causing this finding, we would expect consistent hypo or hypermethylation across all regions (as the array-specific error should not be specific to any particular region). As evidenced in Fig. 2c-d, variation in the open sea region is greater than that of the island regions, however given the large sample size of probes in open sea regions (228,222 CpGs compared to 76,982 island CpGs as noted in Table 4), we feel this finding is robust.

5. *Case characteristics/Table 1: There appears to be a major bias towards female twin pairs. Is it a sample bias or are cases more prevalent amongst female monozygotic twins? What does the asterisk mean by the p-*

value next to birthweight? Is it possible to provide more granular level of information about the ALL diagnoses? Not otherwise specified (NOS) is not explained nor is it clear why only one patient is indicated to have t(12;21)ETV6-RUNX1 positive ALL, when approximately 20% of cases should have this aberration?

The higher percentage of female twin pairs included in this study is more likely the result of random sampling bias from the registry data, as there is no data in the literature to suggest female twin gestations have a higher risk of pediatric ALL than males. We have now made note of this in the manuscript at lines 132-133. As with singletons, males are at a slightly increased risk of pediatric ALL compared to females; to date, no study has shown variation in this risk specifically in twins or higher order multiples. In general, studies on twin and higher-order multiple births have shown no difference in pediatric ALL risk (see references 1, 2 below), though one recent study conducted in Switzerland indicates an increased risk associated with higher order multiples (reference 3 below). Thus, we have no reason to believe there is a biological explanation for the sex bias in discordant twin cases noted in this study.

The asterisk next to weight on Table 1 was intended to serve as an annotation for the statistical test employed. This notation has been adjusted for clarity to an alternative symbol and referenced in the table legend.

We provided the most detailed level of ALL subtype-specific data provided within the California Cancer Registry (CCR) data. This included a single case designated with a molecular subtype (*ETV6-RUNX1* translocated). Notably, many of the cases included in this study were diagnosed before ALL subtype-specific diagnosis was commonly included in the CCR (post 2010, when the diagnostic codes for ALL subtypes were introduced), which led to the relative lack of detail in this section. This information is unfortunately not retrievable for these subjects.

References:

1. Puumala SE, Carozza SE, Chow EJ, Fox EE, Horel S, Johnson KJ, et al. Childhood cancer among twins and higher order multiples. *Cancer Epidemiol Biomarkers Prev.* 2009;18(1):162-8.
2. Murphy MF, Bunch KJ, Chen B, Hemminki K. Reduced occurrence of childhood cancer in twins compared to singletons: protection but by what mechanism? *Pediatr Blood Cancer.* 2008;51(1):62-5.
3. Lupatsch JE, Kreis C, Konstantinoudis G, Ansari M, Kuehni CE, Spycher BD. Birth characteristics and childhood leukemia in Switzerland: a register-based case-control study. *Cancer Causes Control.* 2021;32(7):713-23.

6. A general complaint about the manuscript is the extensive focus on gene enrichment and genomic location analyses. These analyses are indeed valuable, but are too lengthy in the main text/figures and the paper could be made sharper if much of those analyses were moved to the supplement and only the main take home points would be presented in the main text.

To better focus attention to the main results of this study, we have moved Fig. 3 and Fig. 4 from the main paper body to the supplementary section and reduced the total figures number to 2. We have additionally removed supplementary tables referring to repetitive element names and families, and limited discussion of these two items in the results section. We feel this better focuses the results on the most important take home points.

7. The figures can be generally improved to be more informative and easier to read. Specifically, for Figure 2: the plots are too large (with excessive white space) and text size on the axes too small, making them hard to read. Some of the results would be better suited in a table (panels A & B for example). The box plots (C & E) would be improved with actual median data points. The figure legend says panel C contains the distribution of median delta beta-values but the box plot doesn't do a good job showing the distribution of these data points. Perhaps the actual distribution of delta beta values (not the median) with a violin plot or similar would do a better job depicting this. Panels D and F are lacking explanation of the color scheme, which seems a bit random. The large overlap in data points could be made clearer by adding jitter to separate the overlapping points.

Figures have been adjusted for improved clarity. Specifically, we added gridlines to plots to provide better visualization of the data and increased font size for all text (see Fig. 1 and 2 for adjustments). We have added median data points to plots in Fig. 2c and 2d to accurately demonstrate the distribution of data in a clear manner for the reader. The color scheme for Fig. 2e-h is randomly assigned by twin pair. The 30 pair included in Fig. 2e and Fig. 2g are matched by color on the two plots. The other 11 pair depicted in Fig. 2f and 2h are also matched by color. We attempted to use jitter to better separate data points for Fig. 2e-h, however this made assessment of pairings (represented by the lines on the plot) less interpretable.

We additionally adjusted supplementary figures for improved clarity by adding gridlines to Supplementary Fig. 1, 6, 8, and 9. Supplementary Fig. 6 (Illustrating the significant DMR at *TRIM39-RPP21*) has additionally been revised to more clearly depict regression results (coefficients and P-values) as well as median delta beta distribution (to indicate the paired distribution of DNA methylation in cases and controls) for each of the 9 CpGs within the region. Finally, Supplementary Fig. 3 (from the initial submission) has been removed due to redundancy.

8. How/why were a total of 37 twin pairs included in the conditional regression analysis for B-ALL when in table 1 it is stated that 32 cases are B-cell, 4 are T-cell and the remaining 7 were not listed or unknown? The decision to remove the four cases who later develop T-ALL is perhaps a little confusing given that it seems that the some of the cases with unknown cell lineage were included in the regression analysis anyways.

For the conditional regression analysis, we aimed to evaluate subjects of the most similar assumed underlying biology regarding future development of ALL. Given limitations in subtype-specific diagnostic information, we felt this was best accomplished by focusing on cell lineage subgroups. For this reason, we excluded the 4 T-cell lineage cases from this analysis. While we intended to assess these individuals in an independent analysis, the small sample size (n = 4 pair, or 8 individuals) was not sufficient for model fitting. We chose to include the 7 "unknown" lineage cases in our analysis to balance the goals of maintaining sufficient sample size to accommodate the conditional regression model as well as evaluating the most similar assumed underlying biology. Given T-cell cases in general make up a smaller percentage of ALL diagnoses, we would assume a large proportion of the unknown cases are B-cell lineage ALL. We have now made notation of this at line 175.

We ran the conditional regression model using both the full cohort (n = 41 twin pairs) and known B-cell cases only (n = 30) to assess the sensitivity of our results to the inclusion or exclusion of these subjects. While the full cohort analysis (n = 41) resulted in fewer significant DMPs (192 probes with FDR < 0.05), the significant bias toward negative coefficients unchanged both globally and within the regions described in the manuscript. For known B-cell ALL cases one (n = 30 pair), the number of significant DMPs was increased to 513 with FDR < 0.05, however again

with a similar distribution of coefficient directions by region. The difference between open sea and island regions is more distinct in this cohort, as island probes demonstrate a slight positive bias in coefficients whereas open sea coefficients remain significantly negative. Notably, Q-Q plots show genomic inflation is elevated in comparison to the model shown in the manuscript (lambda values 1.12 and 1.06 versus 1.02 in the manuscript model. Please see the figures below which detail these findings, comparable to Fig. 1b and Fig. 2b in the manuscript. We have made note of this analysis in the manuscript at lines 178-180.

9. The manuscript is lacking a discussion/analysis of time to ALL diagnosis and magnitude of global DNA hypomethylation. On lines 251-253, it was noted that the 30 twin pairs that demonstrated a “universal hypomethylation” didn’t differ in terms of some phenotypes, but it was not mentioned if they differed in time to ALL diagnosis, which would have been interesting to look at. Importantly, processing batch of bisulfite conversion/location on chip or other technical reason for lower DNA methylation level should be looked at.

We compared available registry data between the 30 twin pairs demonstrating global hypomethylation in cases, including assessment for association with sex, birthweight, age of leukemia diagnosis, diagnosis code, and array chip/batch number. None of these factors which

might differ between twin siblings were significantly different in hypomethylated cases compared to twin pairs with case DNA hypermethylation. We have further noted this in the manuscript discussion section at lines 359-361.

10. It is further unclear why a simple “global” assessment of DNA methylation % was not calculated and compared within a pair and also between individuals/twin pairs? Twins are assumed to show more similar DNA methylation to one another than in comparison to unrelated individuals, is this the case in this dataset?

We have amended Fig. 2 to illustrate a comparison of global DNA methylation content between cases and controls (please see Fig. 2a). Specifically, we compared median global DNA methylation content in a paired non-parametric test to show significant global hypomethylation in ALL cases ($P = 0.048$). We have further amended the manuscript to specifically note this result in lines 228-229 and have added Supplementary Table 13 listing median global DNA methylation values for cases and controls by twin pairing.

We find twins in this study show more similar DNA methylation to one another than to other unrelated individuals. As noted in the response to comment #7 above, we added Supplementary Fig. 2-4 to provide a comparison of DNA methylation correlation in twins and unrelated individuals in the dataset. Notably, while correlation across the full array is high for both twins and unrelated individuals ($R = 0.968-0.991$ in twins, $R = 0.964-0.984$ in unrelated individuals), the twin pairs remain more highly correlated ($R = 0.900-0.990$) in the top 1000 most variable CpG sites while unrelated individuals show far lower correlation ($R = 0.596-0.675$).

11. Given the rarity and difficulty in obtaining these sample types, validation on external samples is understandably difficult. However, some level of technical validation of some of the findings would be required at a minimum, to make sure that these observations are technically reproducible and not due to an unaddressed batch effect. Especially given the generally small DNA methylation differences within the twins observed.

A means to validate results, we conducted DNA methylation-specific droplet digital PCR (ddPCR) on a set of significant DMPs identified in the regression analysis. Details of this new analysis are included in the methods section (lines 574-606), with a description of results in lines 205-223 and Supplementary Fig. 7. Additional supporting tables for this analysis are shown in Supplementary Tables 11 (ddPCR results by twin pair), 12 (a comparison of ddPCR and array-based results), and 18 (a description of ddPCR primer and probe design). Briefly, we selected twin pairs with sufficient remaining sample for additional analysis in both the case and control sibling ($n = 9$ pairs, 18 individuals). We then selected 4 significant differentially methylated probes from the conditional regression analysis ranked by the highest intrapair variability in DNA methylation results. These sites included probes at *TRIM39*, *SDHC*, *CMIP*, and *FOXK1*. Please see the methods section referenced above for further details into the design of this assay.

We identified significant correlation in DNA methylation when comparing normalized results from the array-based approach (beta values) and ddPCR (fractional abundance) across all individuals and probes as represented in Supplementary Fig. 7a ($R = 0.81$, $P < 2.2 \times 10^{-16}$). Significant correlation was additionally identified across individuals for each of the 4 individual gene targets (Supplementary Fig. 7a).

To assess correlation in biological effect, we next compared normalized delta-DNA methylation values (delta-beta values for array-based data, and delta-fractional abundance scores for

ddPCR data). Correlation remains significant across the two methods ($R = 0.59$, $P = 0.00013$) for all gene targets combined. Individually, we identified stronger correlation for *TRIM39* ($R = 0.68$) and *FOXK1* ($R = 0.66$) compared to *CMIP* and *SDHC* ($R = 0.52$ for both), though all 4 targets demonstrate positive correlations (Supplementary Fig. 7b).

These results demonstrate that DNA methylation results are correlated across the two methods. Furthermore, when assessing the biological finding of interest (the delta-DNA methylation values), we find this correlation in methods is maintained. The ddPCR analysis utilized a separate isolation and bisulfite treatment of genomic DNA samples from these twin pairs, thus increasing our confidence that the array-based results represent true biological phenomena rather than technical variation related to the array or bisulfite conversion.

12. *The references in the introduction could use updating. Many are 10-20 years old and there are many more up to date references that would be suitable for describing the current situation in the ALL field.*

We have updated the references to reflect more up to date articles. Specifically, see reference numbers 1, 2, and 6.

13. *How do the authors see their results in comparison to other similar studies in other pediatric hematological malignancies (ie. Behnert et al, 2021 Leukemia, JMML, for example).*

We appreciate the reviewer's suggestion to compare our results to similar studies in pediatric hematological malignancies. To our knowledge, no other study in pediatric lymphoid malignancies has evaluated pre-diagnostic samples to identify significant associations with future cancer development. For this reason, we cited studies evaluating DNA methylation variation in ALL blast cells at the time of diagnosis (manuscript references 14, 15, and 18) as a means of comparison for our results. We additionally reference an investigation of epigenetic remodeling in B-ALL (manuscript reference 17), which describes pathways of DNA methylation reprogramming in leukemic cells compared to normal pre-B cells. Notably, this study identified backbone regions of the genome, including open sea regions, are primarily demethylated in B-ALL leukemic cells compared to normal pre-B cells which is consistent with the results of the current manuscript.

The interesting study suggested here (Behnert *et al.*) evaluates DNA methylation at birth in 35 future JMML cases and 12 controls using a custom-capture, targeted MethylSeq approach to evaluate 1,386 CpG sites. Results from these subjects were compared to JMML-specific DNA methylation subgroups (described in the reference listed below) identified using the same MethylSeq approach. All newborn samples assessed, including all cases and controls, fell into the "low methylation (LM)," categorization for JMML. The authors noted the DNA methylation status was similar between cases and controls in the study.

The small number of CpG sites assessed and small sample size of genetically dissimilar individuals makes comparison between Behnert *et al.* and the current manuscript difficult. While the 1,386 CpGs are sufficient for categorization according to JMML subgroups, we feel these findings are not adequately representative of genome-wide DNA methylation status. For this reason, we did not include this particular study in our manuscript despite its assessment of pre-diagnostic DNA methylation status in a pediatric hematologic malignancy.

Reference:

Schönung M, Meyer J, Nöllke P, Olshen AB, Hartmann M, Murakami N, Wakamatsu M, Okuno Y, Plass C, Loh ML, Niemeyer CM, Muramatsu H, Flotho C, Stieglitz E, Lipka DB. International Consensus Definition of DNA Methylation Subgroups in Juvenile Myelomonocytic Leukemia. *Clin Cancer Res.* 2021 Jan 1;27(1):158-168. doi: 10.1158/1078-0432.CCR-20-3184. Epub 2020 Nov 2. PMID: 33139265; PMCID: PMC7785676.

REVIEWER COMMENTS

Reviewer #1 (Remarks to the Author):

Thank you for addressing all my remarks. I am satisfied with your responses.

Reviewer #2 (Remarks to the Author):

The revised manuscript by Nickels et al adequately answered most of the questions. Upon reading the revised version one critique remains and couple additional questions are raised:

1. As pointed out previously (question 2, reviewer 2), a PCA plot (or hierarchical clustering) across all CpG beta values & labeled by the 41 twin pairs, location on array, array batch, etc, etc, would be help demonstrate that there are not major batch effects.
2. The within-twin pair correlation coefficients should be listed per pair in a supplemental table. Or, even preferably, in a correlation matrix where the pairwise correlations between all the samples in the cohort are given. In this way, the authors can compare the within twin to between individual correlations more comprehensively. Giving only the correlations of 5 randomly matched individuals (lines 146-152)is insufficient.
3. The high correlation (within twin pair) and lower correlation (between unrelated individuals) analysis for the 1000 most variable CpG sites is not surprising. Isn't it likely that the influence of genetic-background cause variability in the DNA methylation values when the dataset is analyzed in this way? If the authors wish to include this analysis, the reason behind the analysis and result should more clearly justified.
4. As stated on lines 141-144, Cell proportions were compared across twin pairs between cases and unaffected siblings ... which showed no significant differences in nucleated cell proportions (paired Wilcoxon signed rank test). However, the results of this analysis are presented the group level (cases vs controls) and several outliers are seen in supplementary Figure S1. The figure would be much more informative if there was a connection between the paired samples, like in Figure 2A. The values should be provided for each sample/pair individually in the supplemental information.
5. Figure 2a, two twin pairs have much lower median global DNA methylation content, of which one pair has 15% lower global median methylation than the others. Which pairs were these and is there any explanation for the drastic difference? Is the twin pair with much lower median methylation the same that also appears to be an outlier in Supplemental figure 5, at least for 5 of the CpG sites?
6. A paper recently published focusing on genome-wide DNA methylation analysis in ALL (<https://doi.org/10.1038/s43018-022-00370-5>) provides interesting insight into the global DNA methylation question in ALL. Please refer to this paper and include it in the discussion on global methylation status of ALL genomes on lines 367-375.

July 1, 2022

Response to Reviewer Comments (Second Review) [NCOMMS-21-39756B]

Dear Editorial Board and Reviewers,

We extend our gratitude to the reviewers for their positive comments. Below, we provide detailed responses to their specific concerns and commentary. We feel these revisions have further strengthened our manuscript, "DNA Methylation at Birth in Monozygotic Twins Discordant for Pediatric Acute Lymphoblastic Leukemia." The revised manuscript includes additional analysis, with specific emphasis focused on evaluating raw data for evidence of underlying systematic bias or technical artifact which might obfuscate the reported results. More specifically, we conducted tSNE analysis and a pairwise correlation matrix evaluating site specific DNA methylation correlation across all individuals included in the study. The results of these new analyses are described in detail below and are referenced in the revised manuscript. These additional analyses augment our confidence in the biological relevance and veracity of the reported results and add further clarity as to the underlying data structure.

Reviewer #1 (Remarks to the Author)

Thank you for addressing all my remarks. I am satisfied with your responses.

We thank Reviewer #1 for your critical review and important commentary regarding this manuscript.

Reviewer #2 (Remarks to the Author)

The revised manuscript by Nickels et al adequately answered most of the questions. Upon reading the revised version one critique remains and couple additional questions are raised:

1. As pointed out previously (question 2, reviewer 2), a PCA plot (or hierarchical clustering) across all CpG beta values & labeled by the 41 twin pairs, location on array, array batch, etc, etc, would be help demonstrate that there are not major batch effects.

We thank the reviewer for this suggestion. To evaluate for potential bias due to batch effects or technical artifact from the EPIC array, we conducted tSNE analysis on beta values from 695,997 CpG probes (excluding chromosomes X and Y) using the R package "RTsne." Results from this analysis are presented as Supplementary Fig. 3 in the revised manuscript, along with a brief discussion regarding this analysis at lines 146 – 148. Notably, the tSNE plot organizes the data by twin pair units (as noted by the labels on the plot), however there is no obvious organization by array chip used in obtaining the DNA methylation data. When analyzing the full set of CpG probes evaluated in the manuscript (710,010 CpGs, including chromosomes X and Y), the data organized by male and female sex, as well as twin pairings, however no clear association was seen with array Chip. We feel this offers evidence of a lack of underlying batch effect influencing DNA methylation values, and further strengthens our confidence in the validity of our reported results.

To further address this question, we additionally conducted both principal component analysis and hierarchical clustering of the data. As demonstrated in the PCA plot provided below, there is no appreciable association with array chip, which we feel further emphasizes the lack of batch effect impacting the reported results. Notably, both individuals from a single twin pair (pair 14, as indicated below) clustered separately from the group, largely based on the first principal component vector. Hierarchical clustering, as demonstrating on the heatmap provided below, further demonstrates lack of evidence for notable batch effect or technical variation in the data.

PCA plot of beta values for 695,997 CpG probes (chromosomes X and Y omitted) for the 41 twin pairs (82 individuals) included in this study.

Heatmap demonstrating hierarchical clustering for beta values from 695,997 CpG probes (omitting chromosomes X and Y) for 41 twin pairs (82 individuals) included in this study. Heatmap legend includes random pair ID (which is additionally designated by twin A and B along the X-axis), sex, ALL-case status, and DNA methylation array chip.

Most individuals clustered strongly by twin pair identity, with notable exceptions. Specifically, both individuals from twin pair 14 again clustered separately from the data, constituting the initial branch point in the dendrogram. As is the case with all paired twins, twin pair 14 were block randomized to separate chips thus we cannot point towards a technical artifact impacting a single chip for this result. Further investigation revealed both individuals in twin pair 14 have the lowest degree of global DNA methylation (as demonstrated in manuscript Figure 2a), and constituted outliers regarding nucleated red blood cell (nRBC) proportion as evidenced in the deconvolution analysis (see response to comment #4 below; pair 14 is the only pair with an nRBC proportion greater than 0.75). Notably, birth records indicate a pregestational diagnosis of diabetes for the mother of twin pair 14, a diagnosis absent in all other individuals included in the study. Pregestational diabetes and chronic intrauterine hypoxia are associated with increased nRBCs in offspring at birthⁱ. While the exact cause of the elevated nRBC level for this twin pair cannot be determined, we suspect the maternal diagnosis of pregestational diabetes may have contributed to this finding. Importantly, our conditional regression model included nucleated cell proportions derived from deconvolution analysis, including nRBC level. We thus feel the impact of this variation is adequately controlled in the conditional regression analyses in our study.

In addition, a minority of individuals did not cluster together with their related twin sibling. When repeating the hierarchical clustering analysis with all array probes (710,010; including chromosomes X and Y), siblings twin pairs 21 and 51 clustered separately. There are no obvious differences in cell proportions for these pairs to explain this difference, however both twin pairs had blood spots collected at a wider time interval than most individuals included in the study at 19 days for pair 21 and six days for pair 51. No other collection interval exceeded 24 hours for the remaining twin pairs.

Citations for this comment:

i. Hermansen, M. C. Nucleated red blood cells in the fetus and newborn. *Arch Dis Child Fetal Neonatal Ed* **84**, F211-215 (2001). <https://doi.org:10.1136/fn.84.3.f211>

2. The within-twin pair correlation coefficients should be listed per pair in a supplemental table. Or, even preferably, in a correlation matrix where the pairwise correlations between all the samples in the cohort are given. In this way, the authors can compare the within twin to between individual correlations more comprehensively. Giving only the correlations of 5 randomly matched individuals (lines 146-152) is insufficient.

To address this concern, we have created a correlation matrix demonstrating pairwise correlations across all individuals included in the study. Furthermore, to visualize this data we have provided the correlation matrix with hierarchical clustering as Supplementary Fig. 2 in the revised submission. With this addition, we have omitted the prior data regarding the 5 randomly matched individuals at line 146 – 148 and have removed Supplementary Fig. 4 from the prior submission. As noted in response to comment #1, twin pair 14 shows high intrapair correlation, however both individuals in this pair are less correlated to unrelated individuals in the study. While the majority of individuals show highest correlation with their related twin sibling, a number cluster with random individuals on the plot. As noted in the response to comment #1, this is partially explained by variation in the time interval in which blood spots were obtained from individuals following birth and may be further explained by subtle (though not statistically significant) variation in nucleated cell proportions, which are not controlled for in this clustering algorithm.

3. The high correlation (within twin pair) and lower correlation (between unrelated individuals) analysis for the 1000 most variable CpG sites is not surprising. Isn't it likely that the influence of genetic-background cause variability in the DNA methylation values when the dataset is analyzed in this way? If the authors wish to include this analysis, the reason behind the analysis and result should more clearly justified.

We thank the reviewer for pointing out this concern and agree that the 1000 most variable CpG sites are likely highly influenced by genetic contribution the DNA methylation status. Considering the additional analysis and results provided in response to comments #1 and #2, we have now omitted Supplementary Figs. 2, 3, and 4 from the prior edition of the manuscript. We feel the addition of new Supplementary Figs. 2 and 3, along with the provision of the correlation matrix data as a supplementary data file, more adequately addresses the original concern regarding correlation in DNA methylation beta values between twin pairs.

4. As stated on lines 141-144, Cell proportions were compared across twin pairs between cases and unaffected siblings ... which showed no significant differences in nucleated cell proportions (paired Wilcoxon signed rank test). However, the results of this analysis are presented the group level (cases vs controls) and several outliers are seen in supplementary Figure S1. The figure would be much more informative if there was a connection between the paired samples, like in Figure 2A. The values should be provided for each sample/pair individually in the supplemental information.

We have revised Supplementary Fig. 1 to address these comments. Specifically, we have added connections on the plot to highlight twin pairings for each nucleated cell type. In general, twin pairs demonstrated similar nucleated cell proportions for all cell types. As demonstrated in the revised figure, outliers also tended to occur by twin pair groups, rather than widely variable values across twins within an individual related pair. We have additionally provided individual level deconvolution data for twin pairings in the source data files accompanying this submission.

5. Figure 2a, two twin pairs have much lower median global DNA methylation content, of which one pair has 15% lower global median methylation than the others. Which pairs were these and is there any explanation for the drastic difference? Is the twin pair with much lower median methylation the same that also appears to be an outlier in Supplemental figure 5, at least for 5 of the CpG sites?

As noted in response to comment #1, the outliers in Figure 2a displaying lower median global DNA methylation content represent both individuals from twin pair 14. Similarly, twin pair 14 does indeed appear as the outlier with lower DNA methylation status in various CpG sites for Supplementary Fig. 5. We feel the most likely explanation for this variation is the substantially higher proportion of nRBCs identified in deconvolution analysis for both siblings in twin pair 14 compared to all other subjects, as this cell type is known to possess a hypomethylated epigenetic profile¹. As noted in Supplementary Table 13, the median global DNA methylation content for the case sibling is higher than that of the control sibling in twin pair 14, thus despite the above noted findings this pair did not contribute to the general trend of ALL-associated global hypomethylation described in the manuscript. Importantly, as nucleated cell proportions are included in the deconvolution analysis described in the manuscript, we feel we have provided adequate control for this variation in our analytical approach. We have added notation regarding this specific twin pair to the manuscript text at lines 238 – 244.

Citations for this comment:

i. de Goede OM, Lavoie PM, Robinson WP. Characterizing the hypomethylated DNA methylation profile of nucleated red blood cells from cord blood. *Epigenomics*. 2016;8(11):1481-94.

6. A paper recently published focusing on genome-wide DNA methylation analysis in ALL (<https://doi.org/10.1038/s43018-022-00370-5>) provides interesting insight into the global DNA methylation question in ALL. Please refer to this paper and include it in the discussion on global methylation status of ALL genomes on lines 367-375.

We thank the reviewer for calling attention to this important, recently published work. Hetzel, *et al.* describe DNA methylation profiles of 82 ALL samples (including Ph-Like, *DUX4*-rearranged, and hypodiploid B-ALL subtypes, as well as T-ALL cases) compared to healthy B and T-cell precursor cells using whole genome bisulfite sequencing. The authors conclude the global DNA methylation landscape of ALL differs from the canonical cancer methylome in that global hypomethylation does not feature prominently in this malignancy. Localized regions of hypermethylation remains consistent in ALL in their study. As demonstrated in Fig. 1d in the paper, both B-ALL and T-ALL remain highly methylated in comparison to solid tumors, chronic lymphocytic leukemia, and mantle cell lymphoma. Importantly, while T-ALL remains nearly equal in global methylation to healthy progenitor cells, a subtle but notable decrease in global DNA methylation is noted for all three B-ALL subtypes assessed, as well as a separate group of B-ALL samples of unknown subtype (“Blueprint” samples as noted in the figure). While these B-ALL samples do not demonstrate a high degree of global hypomethylation consistent with the solid tumors evaluated in their study, the more subtle shift toward hypomethylation is consistent with the findings reported in our manuscript. We find these results to further support a subtle but consistent pattern of decreased global DNA methylation in B-ALL. We have added commentary regarding this reference in the discussion section of the revised manuscript at lines 389 – 394.

The results provided by Hetzel *et al.* are limited in some degree due to the selected nature of ALL subtypes evaluated. While the study did evaluate a group of B-ALL cases of unknown subtype (the Blueprint cases noted above), their study focused primarily on three ALL subtypes (Ph-like, *DUX4*-rearranged and hypodiploid) which had not been specifically investigated for global DNA methylation changes in prior investigations. Thus, the results are likely not universally representative of the molecular landscape of pediatric B-ALL. This limits direct comparisons with the results of our investigation, which primarily included cases with unknown molecular subtypes.

REVIEWERS' COMMENTS

Reviewer #2 (Remarks to the Author):

Thank you for comprehensively addressing all my comments and I am satisfied with your responses.

September 5, 2022

Response to Reviewer Comments (Third Review) [NCOMMS-21-39756C]

Dear Editorial Board and Reviewers,

We again would like to extend our gratitude to the reviewers for their positive comments and support of our manuscript, "DNA Methylation at Birth in Monozygotic Twins Discordant for Pediatric Acute Lymphoblastic Leukemia." These detailed and thoughtful comments have guided revisions of the work and have undoubtedly strengthened the manuscript. At this time, there are no additional specific comments to address in this response to reviewers.

Reviewers' Comments

Reviewer #2 (Remarks to the Author):

Thank you for comprehensively addressing all my comments and I am satisfied with your responses.